# Rieske iron-sulfur protein induces FKBP12.6/RyR2 complex remodeling and subsequent pulmonary hypertension through NF-κB/cyclin D1 pathway

Lin Mei[1,4], Yun-Min Zheng[1,4], Tengyao Song[1], Vishal R. Yadav[1], Leroy C. Joseph[1], Lillian Truong[1], Sharath Kandhi [1], Margarida M. Barroso [1], Hiroshi Takeshima[2], Marc A. Judson[3] & Yong-Xiao Wang [1✉]

Ca$^{2+}$ signaling in pulmonary arterial smooth muscle cells (PASMCs) plays an important role in pulmonary hypertension (PH). However, the underlying specific ion channel mechanisms remain largely unknown. Here, we report ryanodine receptor (RyR) channel activity and Ca$^{2+}$ release both are increased, and association of RyR2 by FK506 binding protein 12.6 (FKBP12.6) is decreased in PASMCs from mice with chronic hypoxia (CH)-induced PH. Smooth muscle cell (SMC)-specific RyR2 knockout (KO) or Rieske iron-sulfur protein (RISP) knockdown inhibits the altered Ca$^{2+}$ signaling, increased nuclear factor (NF)-κB/cyclin D1 activation and cell proliferation, and CH-induced PH in mice. FKBP12.6 KO or FK506 treatment enhances CH-induced PH, while S107 (a specific stabilizer of RyR2/FKBP12.6 complex) produces an opposite effect. In conclusion, CH causes RISP-dependent ROS generation and FKBP12.6/RyR2 dissociation, leading to PH. RISP inhibition, RyR2/FKBP12.6 complex stabilization and Ca$^{2+}$ release blockade may be potentially beneficial for the treatment of PH.

[1] Department of Molecular and Cellular Physiology, Albany Medical College, Albany, 12208 NY, USA. [2] Department of Biological Chemistry, Kyoto University Graduate School of Pharmaceutical Sciences, Kyoto, Japan. [3] Division of Pulmonary and Critical Care Medicine, Albany Medical College, Albany, 12208 NY, USA. [4] These authors contributed equally: Lin Mei, Yun-Min Zheng. ✉email: WangY@amc.edu

Pulmonary hypertension (PH) is a common disease with a short median survival time after diagnosis mostly because of limited specific and effective therapeutic options[1]. This devastating condition often occurs in respiratory illnesses (e.g., chronic obstructive pulmonary disease) and high-altitude residence due primarily to chronic hypoxia (CH)-induced pulmonary artery (PA) constriction and remodeling. These two major cellular responses are principally mediated by an increase in $[Ca^{2+}]_i$ in PA smooth muscle cells (PASMCs)[2–4]. However, the underlying mechanisms of the increased $[Ca^{2+}]_i$ and associated PH remain largely uncertain.

$Ca^{2+}$ release from the sarcoplasmic reticulum (SR) through ryanodine receptors (RyRs) is proven to be one of the major routes for the increased $[Ca^{2+}]_i$ in PASMCs, and all three subtypes of RyRs (RyR1, RyR2, and RyR3) are expressed in this type of cells[5,6]. We have shown that RyR2-mediated $Ca^{2+}$ release serves as an instrumental process in hypoxia-evoked contraction and remodeling[7–9], which are linked to canonical transient receptor potential (TRPC) channel activation and voltage-dependent $K^+$ (Kv) channel inhibition[10]. These data suggest that RyRs are integral to the development of hypoxic cellular responses in PASMCs. However, whether and how RyR2-mediated $Ca^{2+}$ signaling exerts its role in the development of PH and the underlying molecular mechanisms are unknown.

FK506 binding protein 12.6 (FKBP12.6), an endogenous RyR2 stabilizer, is associated with RyR2 and inhibits $Ca^{2+}$ release from SR in PASMCs[11,12]. We have further found that acute hypoxia can induce generation of reactive oxygen species (ROS) and oxidize RyR2/FKBP12.6 complex, leading to their dissociation to increase $[Ca^{2+}]_i$ in PASMCs[8]. It is reported that the dissociation of FKBP12.6 from RyR2 can cause $Ca^{2+}$ leak from the SR, which may mediate cardiac diseases[13,14]. Stabilization of RyR2/FKBP12.6 complex with S107 (a 1,4-benzothiazepine derivative) has been proved to prevent ventricular hypertrophy[13], aging-related muscle weakness[15], and type-2 diabetes[16]. However, other reports are debating the role of FKBP12.6 in cardiomyocyte[17–20]. To our best knowledge, there is no exploration of the role of RyRs in the development of PH.

Rieske iron-sulfur protein (RISP) is a major functional component of mitochondrial complex III, which serves as a critical molecule in the initiation and progression of the hypoxic ROS generation. We have demonstrated that RISP knockdown (KD) abolishes the hypoxic ROS formation in isolated PASMCs, whereas RISP overexpression produces an opposite effect. RISP KD also inhibits the hypoxic increase in $[Ca^{2+}]_i$ in PASMCs and HPV in isolated pulmonary arteries[21,22]. The report by Waypa et al.[23] provides further in vivo evidence by using mouse model permitting conditional knockdown of RISP in PASMCs. These mice demonstrate the diminished acute hypoxia-induced ROS production and $[Ca^{2+}]_i$ in pulmonary arteries and attenuated acute hypoxia-induced elevation of right ventricular systolic pressure (RVSP). In the current study, we sought to determine whether RISP knockdown in vivo could affect CH-induced PH as well as how it affects the subsequent altered $Ca^{2+}$ signaling in PASMCs.

Nuclear factor (NF)-κB is a transcriptional factor that is thought to be important in triggering inflammation and cell proliferation. At resting conditions, the NF-κB subunit p50/p65 complex binds to IκB (inhibitory κB), which sequesters this dimer in the cytoplasm. A stimulus may cause degradation of IκB to allow p50/p65 complex to enter the nucleus and elicit a transcription process. Activation of NF-κB has been reported as a potent pathogenetic factor in lung injury, respiratory distress syndrome and PH[24]. Inhibition of NF-κB signal has a therapeutic effect on PH[25]. However, its specific upstream mediators are still elusive, especially in PASMCs. Cyclin D1, one of the key

molecules for G1/S transition, is regulated by NF-κB. It is reported that cyclin D1 was upregulated in PH and associated with $Ca^{2+}$ signaling[26]. Nevertheless, the exact underlying signal cascade remains unidentified.

In this study, we report that CH causes RISP-dependent mitochondrial ROS generation, subsequent FKBP12.6/RyR2 complex dissociation, SR $Ca^{2+}$ leaking, and $[Ca^{2+}]_i$ increase in PASMCs, thereby leading to PA vasoconstriction. The increased $[Ca^{2+}]_i$ also activates NF-κB/cyclinD1 signaling and provokes proliferation in PASMCs and then results in PA remodeling. This cellular response, together with PA vasoconstriction, produces persistent PH.

## Results

**RyR2 gene KO prevents CH-induced $Ca^{2+}$ signaling.** Our in vitro study has shown that acute hypoxia can increase the activity of RyR2 and $Ca^{2+}$ release in PASMCs[8]. Herein, using [$^3$H]-ryanodine binding assay, we found that the maximal binding ($B_{max}$) of ryanodine to RyR was significantly increased from $0.49 \pm 0.06$ (pmol/mg protein) in control PASMCs to $0.7 \pm 0.08$ (pmol/mg protein) in PASMCs from CH mice ($P < 0.05$), as shown in Fig. 1a. The binding affinity of ryanodine to RyR ($K_d$) was decreased from $3.1 \pm 0.5$ in control cells to $1.8 \pm 0.2$ in CH cells. Similar result was observed in human sample as well (Fig. 1b). We also examined RyR $Ca^{2+}$ release in PASMCs from CH and control mice. Application of caffeine, a classic RyR agonist, at a medium concentration (3 mM) showed a much larger increase in $[Ca^{2+}]_i$ in PASMCs from mice exposed to CH than normoxia. The mean $[Ca^{2+}]_i$ increase was $442 \pm 40$ and $800 \pm 72$ nM, respectively (Fig. 1c). However, caffeine (30 mM) to cause a maximal RyR activation induced a decreased $Ca^{2+}$ release in PASMCs from CH mice than control; the mean value was $1040 \pm 30$ nM in the former and $1390 \pm 106$ nM in the latter. These data further suggest that CH induces the increased RyR activity and $Ca^{2+}$ release, but deceases the SR $Ca^{2+}$ store.

To test the role of RyR2, we first compared the ratio of RyR-mediated SR $Ca^{2+}$ leak to SR $Ca^{2+}$ storage in PASMCs from wildtype (WT, RyR2$^{+/+}$) and smooth muscle cell (SMC)-specific RyR2 gene knockout (KO, RyR2$^{-/-}$) mice exposed to normoxia (Nor) and CH. As expected, Western blot and PCR analysis showed the specificity of RyR2 protein KO (Fig. 1d). Direct evidence of SR $Ca^{2+}$ leak was further obtained by using tetracaine (TTC) protocol[14]. The ratio of SR $Ca^{2+}$ leak to SR $Ca^{2+}$ storage was significantly higher in PASMCs from CH than normoxic animals ($5.9 \pm 1.1\%$ vs $15.8 \pm 3.5\%$, $P < 0.05$). Importantly, CH-induced increase in the ratio of RyR-mediated SR $Ca^{2+}$ leak to SR $Ca^{2+}$ storage was fully eliminated in PASMCs from RyR2$^{-/-}$ mice relative to WT (control) animals ($5.3 \pm 1.6\%$ vs $6.5 \pm 1.5\%$), suggesting RyR2 KO was sufficient to reverse the abnormal SR leak in PASMCs from CH mice (Fig. 1e). We also found that store-operated $Ca^{2+}$ entry (SOCE) was largely enhanced in PASMCs from CH mice, and CH-induced SOCE was almost completely inhibited in cells from RyR2$^{-/-}$ mice ($165 \pm 30$ vs $987 \pm 125$ nM, $P < 0.05$) (Fig. 1f). Concurrently, Western blot analysis reveals that TRPC1 and TRPC6 channel protein expression were increased in PASMCs from CH mice, while Kv1.5 channel protein expression levels were decreased. More excitingly, these changes were no longer observed in PASMCs from RyR2$^{-/-}$ mice (Supplementary Fig. 1a–d).

Consistent with the increased RyR activity and $Ca^{2+}$ release, contraction (shortening) of PASMCs was augmented in CH mice compared with normoxic animals ($10.7 \pm 0.2$ vs $9.3 \pm 0.2$ μm, $P < 0.05$). As expected, the increased contraction of PASMCs did not occur in RyR2$^{-/-}$ mice following CH ($11.0 \pm 0.4$ μm vs $11.0 \pm 0.4$ μm, $P > 0.05$, Supplementary Fig. 2a). The resting $[Ca^{2+}]_i$ in

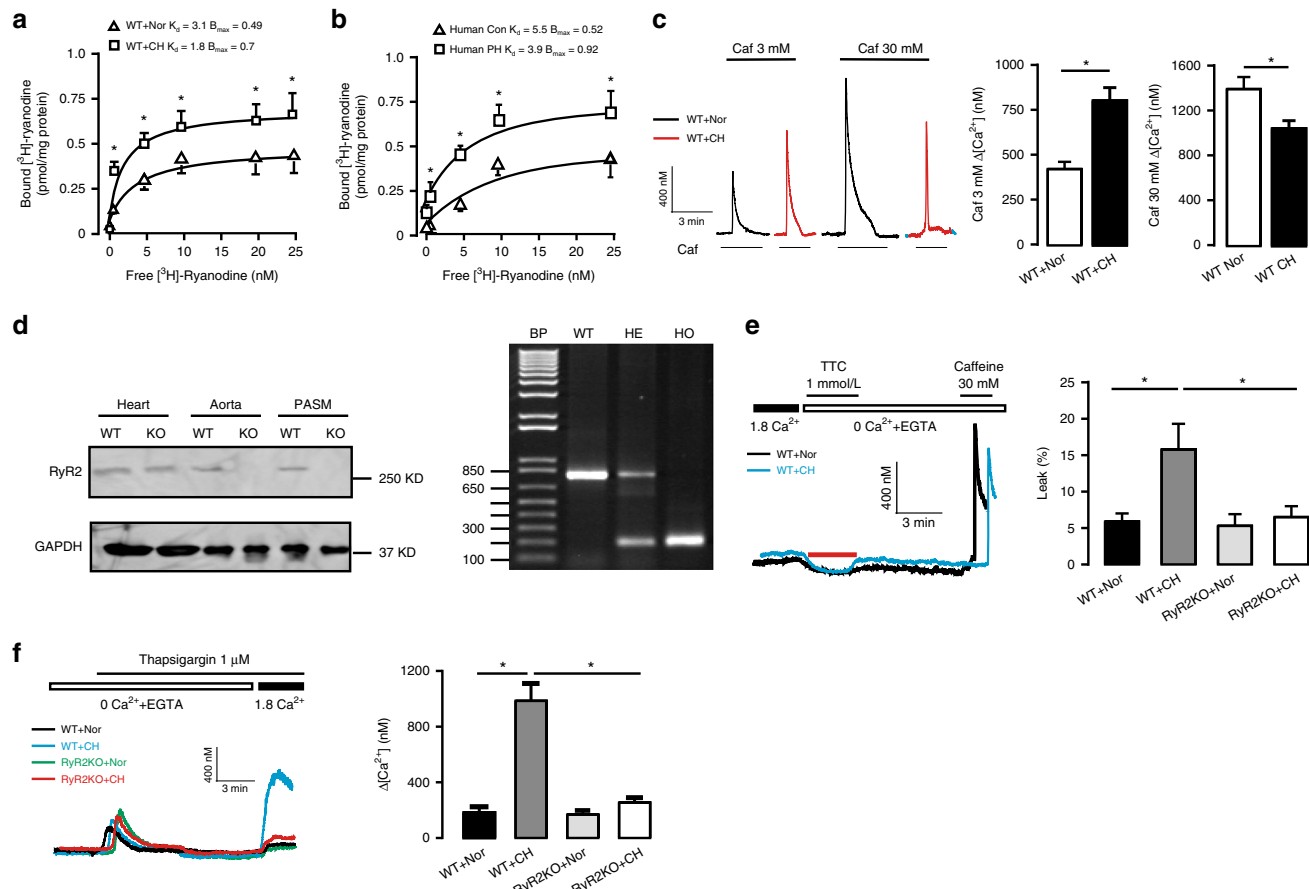

**Fig. 1 Deletion of RyR2 gene prevents CH-induced calcium signal over-activation. a** [³H]-ryanodine binding assays are conducted from wild-type (WT) mice pulmonary arteries (PAs) pretreated with normoxia (Nor) and chronic hypoxia (CH) ($n = 5$ independent studies, 5 mice per study). **b** Human PAs samples with pulmonary hypertension (PH) and age-matched non-PH human PAs (Con) are used to evaluate RyRs activity by [³H]-ryanodine binding assays ($n = 3$ independent studies). **c** Recording of medium (3 mM) and maximal (30 mM) concentration caffeine (Caf)-induced PASMC $Ca^{2+}$ release. Bar graph summary of medium (3 mM) and maximal (30 mM) concentration Caf-induced $Ca^{2+}$ release in WT pulmonary arterial smooth muscle cells (PASMC) after exposure to 3 weeks CH ($n = 7$ independent studies, 95 cells per study). **d** Representative western blot and PCR of RyR2 expression from RyR2 gene knockout (KO) mice. Tissue are harvested from heart, aorta and PASM ($n = 3$ independent studies, 4 mice per study). **e** Representative $[Ca^{2+}]_i$ tracing from Nor and CH PASMC loaded by fura-2/AM switch to 0 $Ca^{2+}$ physiological saline solution (PSS) and 1 mmol/L tetracaine (TTC) to block RyR2. Sarcoplasmic reticulum (SR) $Ca^{2+}$ content is measured by adding 30 mM caffeine. SR $Ca^{2+}$ leak ratio is quantified and normalized to SR $Ca^{2+}$ content in WT and RyR2 KO mice at Nor and CH condition ($n = 4$ independent studies, 50 cells per study). **f** Fura-2/AM $Ca^{2+}$ imaging showing essentially abrogation of store-operated calcium entry (SOCE) activated by thapsigargin (1 μM) in PASMC from RyR2KO mice and statistical results of $Ca^{2+}$ influx peak ($n = 4$ independent studies, 60 cells per study). Data are expressed as mean ± standard error. (*$P < 0.05$, using one-way ANOVA test or Student's $t$ test).

PASMCs was largely increased from 108 ± 10 nM in control mice to 163 ± 11 nM in CH group ($P < 0.05$). The increased resting $[Ca^{2+}]_i$ was fully blocked in RyR2$^{-/-}$ mice (Supplementary Fig. 2b). Therefore, RyR2-mediated SR $Ca^{2+}$ leaking is increased in PASMCs of CH subjects.

Endothelin-1 (ET-1), well-known to be a potent endogenous vasoconstrictor and involved in PH[27], induced a larger $Ca^{2+}$ release in PASMCs from CH mice than normoxic mice; the increases was significantly inhibited in RyR2$^{-/-}$ mice (976 ± 116 vs 570 ± 65 nM, $P < 0.05$, Supplementary Fig. 2c).

**SMC-specific RyR2$^{-/-}$ mice are resistant to CH-induced PH.** We investigated the role of RyR2 in the norepinephrine and endothelin-1(ET-1)-induced contraction of intrapulmonary arteries. As shown in Fig. 2a, CH significantly enhanced contraction in PAs from WT, but not RyR2$^{-/-}$ mice. We also found that CH increased muscularization degree in muscularized

(double elastic lamina > 75%) and partial-muscularized (double elastic lamina < 75%) PAs, but decreased in non-muscularized (only one single elastic lamina) vessels; these effects were blocked in RyR2$^{-/-}$ mice (Fig. 2b). Medial wall thickness (MWT) of PAs in a small (15–50 μm), medium (5–100 μm) and large (>100 μm) size were, respectively, markedly increased from 22 ± 3%, 19.5 ± 3% and 16 ± 3% to 34 ± 4%, 25 ± 3% and 24 ± 3% ($P < 0.05$, Fig. 2c). However, the increased MWT was not found in RyR2$^{-/-}$ mice. Similarly, CH-evoked proliferation of PASMCs, determined by using Ki67 immunostaining, was blocked in RyR2$^{-/-}$ mice (Fig. 2d). RVSP and right ventricular (RV) hypertrophy are the most reliable indicator for PH. Under normoxic conditions, RVSP was unchanged in RyR2$^{-/-}$ mice compared with WT (18.5 ± 1 versus 19.5 ± 2 mmHg, $P < 0.05$). In contrast, CH caused a large elevation in RVSP in WT littermates, but not in RyR2$^{-/-}$ mice (35.5 ± 2.5 versus 21.5 ± 1.5 mmHg) (Fig. 2e). Similarly, CH-induced RV hypertrophy was abolished in RyR2$^{-/-}$ mice (0.31 ± 0.01 versus 0.31 ± 0.01, Fig. 2e). As shown in Supplementary

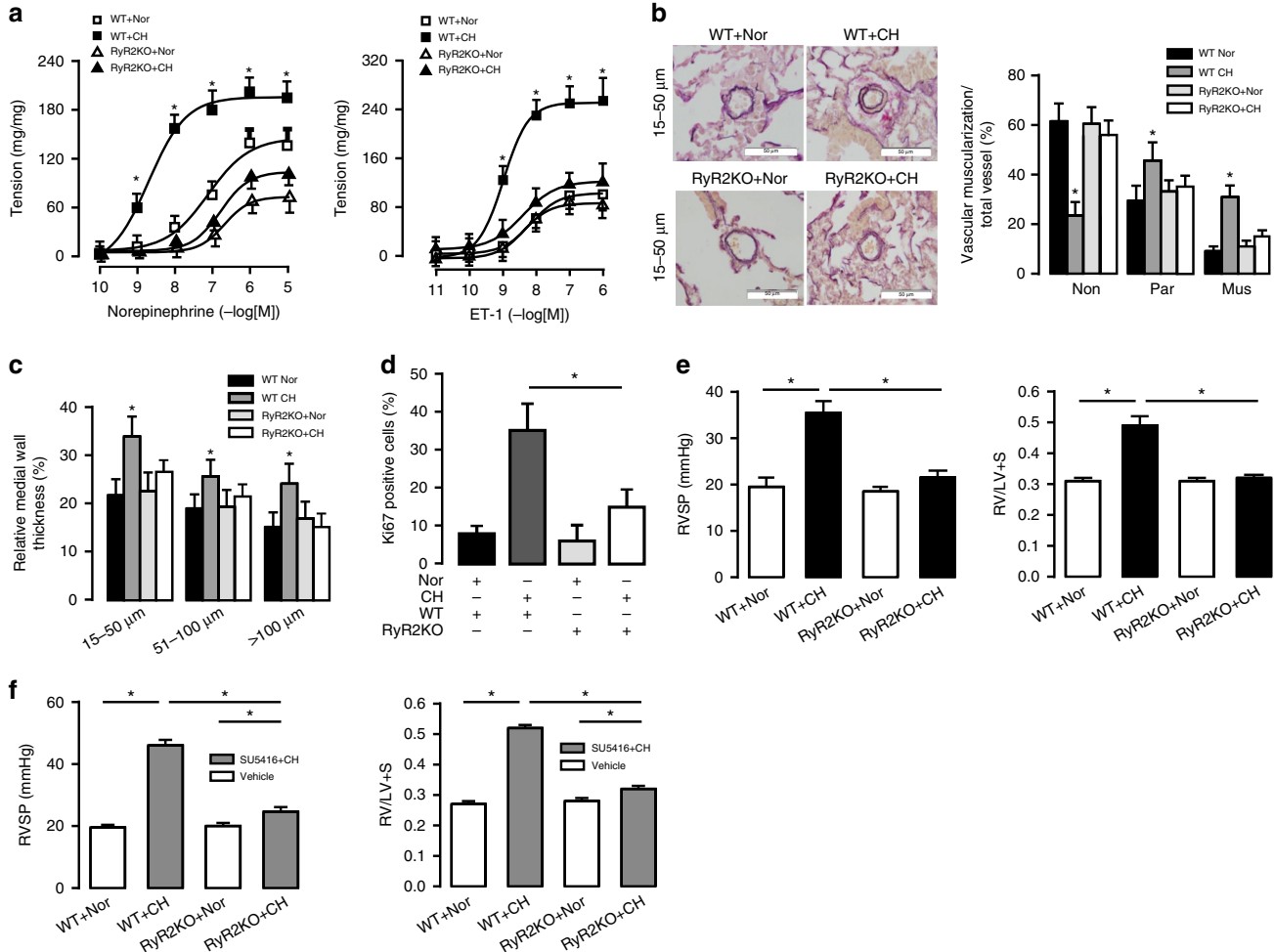

**Fig. 2 RyR2 KO mice are resistant to hypoxia-induced pulmonary hypertension. a** Concentration-response curves for norepinephrine and endothelin-1 (ET-1) induced muscle contraction are obtained from normoxic (Nor) and CH PA strips from WT and RyR2 KO mice ($n = 4$ independent studies, 7 PA strips per study). **b** Typical lung sections of PAs are demonstrated by using Elastic Van Gieson staining in WT mice (bar scale: 50 μm). Proportion of non-(Non), partially (Par), or fully (Mus) muscularized PAs in total counted PAs. A total of 50–60 intra-acinar vessels are analyzed in each mouse, and 5–7 mice have been grouped ($n = 5$ independent studies, 50 vessels per study). **c** Relative medial wall thickness of PAs sized 15–50 μm, 51–100 μm, and over 100μm in relation to cross-sectional diameter ($n = 5$ independent studies, 50 vessels per study). **d** PASMC proliferation (Ki-67 staining) in vivo after CH. The degree of cell proliferation is expressed as the percentage of Ki-67-labeled cells over the total number of cells counted in the media layer of at least 20 muscularized vessels per mouse ($n = 5$ independent studies, 50 vessels per study). **e** Summarized the effect of chronic hypoxia (CH) on the right ventricular systolic pressure (RVSP) in WT and RyR2 KO PH mice model measured by 1.2 F catheter. Ratio of right ventricle (RV) to left ventricle (LV) plus septum (S) weight (RV/LV + S) to measure RV hypertrophy ($n = 6$ independent studies, 5 mice per study). **f** Effect of SU5416 subcutaneously once weekly (20 mg/kg) or vehicle followed by CH on WT RVSP and RV/LV + S ($n = 5$ independent studies, 5 mice per study). Data are expressed as mean ± standard error. (*$P < 0.05$, using one-way ANOVA test).

Fig. 3a, pressure–volume (P-V) measurements showed that the slope of the end-systolic pressure–volume relationships (ESPVRs), an index of contractility independent of preload, became steeper in WT mice following CH, thereby indicating the impaired RV contractility. This CH-induced effect was blocked in RyR2$^{-/-}$ mice (Supplementary Fig. 3b). The RV functional changes were summarized in Supplementary Table 1. We further found that CH/SU5416-evoked increases in RVSP and RV/LV + S were eliminated in RyR2$^{-/-}$ mice (Fig. 2f). All these data demonstrate that RyR2$^{-/-}$ protects mice from CH-induced PH.

**RyR2 inhibition confers protection in CH-induced PH.** Treatment with tetracaine (TTC, a classic RyR antagonist), in mice in vivo significantly attenuated CH-induced increase in $B_{max}$ in SR preparations from PASMCs ($0.34 \pm 0.06$ versus $0.6 \pm 0.1$, $P < 0.05$, Fig. 3a). Caffeine, a classic RyR agonist, induced a larger

increase in $[Ca^{2+}]_i$ in PASMCs from CH mice, and caffeine-induced responses were blocked by in vivo treatment with TTC (Fig. 3b, c). TTC also obliterated CH-evoked PA vasoconstriction (Fig. 3d) and remodeling (Fig. 3e, f). Moreover, we found that after treatment with TTC, CH mice did not develop significantly elevated RVSP ($20.1 \pm 3.8$ versus $24.5 \pm 3.5$ mmHg, $P > 0.05$) and RV/LV + S ($0.3 \pm 0.04$ versus $0.35 \pm 0.04$, $P > 0.05$, Fig. 3g). These results prove that pharmacological inhibition of RyR2 can also protect the development of PH.

**FKBP12.6 is dissociated from RyR2 in CH-induced PH.** Intriguingly, we revealed that RyR2 was mainly expressed in PASMCs, but not in PA endothelial cells (PAECs) (Supplementary Fig. 4). Our previous study has shown that FKBP12.6 is an endogenous RyR2 stabilizer and involved in HPV in vitro[8,11]. As such, we investigated whether FKBP12.6 might mediate the essential role

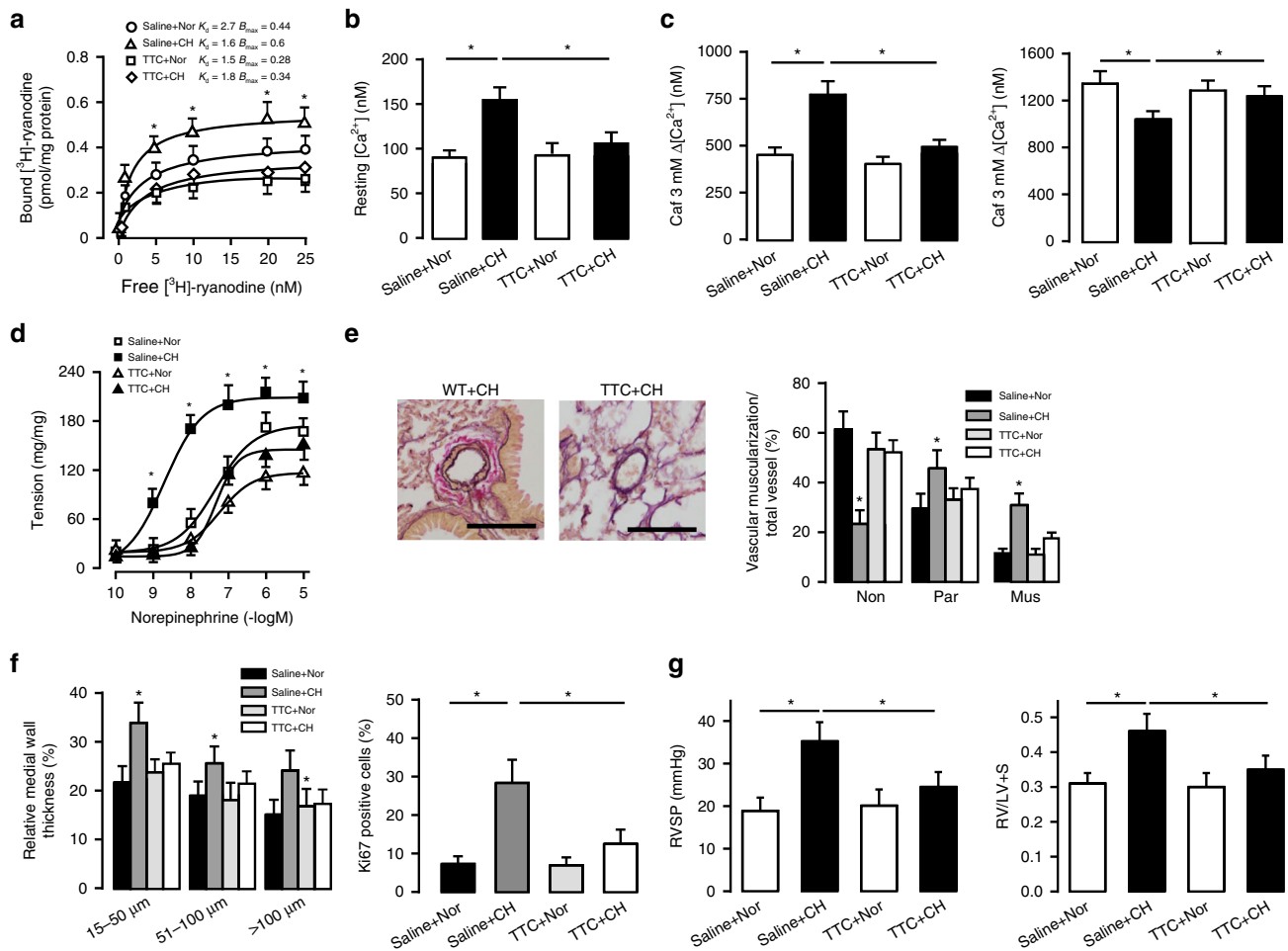

**Fig. 3 Inhibition of RyR2 confers similar protective effect as RyR2 KO in CH-induced PH. a** Determination of RyRs activity by [3H]-ryanodine binding assays are conducted from PAs from TTC treatment group ($n = 5$ independent studies, 5 mice per study). **b** Summary of elevated PASMC resting $[Ca^{2+}]_i$ level after exposure to 3 weeks CH in administration of TTC. PASMC are loaded with fura-2/AM (5 μM) ($n = 5$ independent studies, 60 cells per study). **c** Bar graph summary of medium (3 mM) and maximal (30 mM) dosage of caffeine (Caf) -induced $Ca^{2+}$ peak in TTC treatment group ($n = 5$ independent studies, 60 cells per study). **d** Pulmonary arterial contractility measured by concentration-response curve induced by norepinephrine, which obtained from TTC treated PA strips ($n = 4$ independent studies, 6 PA strips per study). **e** Typical lung sections of PAs from WT mice administrated by TTC (bar scale: 50 μm). Proportion of non-(Non), partially (Par), or fully (Mus) muscularized PAs in total counted PAs ($n = 5$ independent studies, 60 vessels per study). **f** Medial wall thickness of PAs sized 15–50, 51–100, and over 100 μm in relation to cross-sectional diameter ($n = 5$ independent studies, 60 vessels per study). PASMC proliferation (Ki-67 staining) in vivo after 3 weeks hypoxic condition ($n = 5$–7, 50–60 vessels per group). **g** Summarized the effect of CH or Nor on RVSP in saline (control) and tetracaine (TTC, 2 mg/kg/d) treatment group mice model, measured by 1.2F catheter. RV/LV + S to measure RV hypertrophy ($n = 6$ independent studies, 5 mice per study). Data are expressed as mean ± standard error. (*$P < 0.05$, using one-way ANOVA test).

of RyR2 in PH due to the dissociation of these two molecules. Co-immunoprecipitation (Co-IP) found that the ratio of FKBP12.6/RyR2 in SR membrane preparations was decreased by ~4 fold in PASM tissues from CH mice. (Fig. 4a). The similar dissociation of FKBP12.6 from RyR2 also occurred in from patients with PH (Fig. 4b). In support, fluorescence resonance energy transfer (FRET) assay indicated that the energy transfer signal between FKBP12.6 and RyR2 was largely decreased in PASMCs from CH mice (Fig. 4c). We further proved that PASMC proliferation rate was significantly increased in FKBP12.6$^{-/-}$ mice compared with WT animals (30.8 ± 1.7% versus 19.0 ± 2.0%, $P < 0.05$, (Fig. 4d). Furthermore, FKBP12.6$^{-/-}$ mice developed significantly higher RVSP relative to WT mice after CH (42 ± 1.4 mmHg versus 36.3 ± 1.8 mmHg, $P < 0.05$), while there was no difference between FKBP12.6$^{-/-}$ and WT mice under normoxic conditions (19 ± 0.7 mmHg versus 18 ± 1.0 mmHg, $P > 0.05$, Fig. 4e). Consistent with the effect of FKBP12.6 KO, Co-IP study indicated that FK506

administration in mice in vivo (0.05 and 1 mg/kg/d) for 3 weeks dose-dependently promoted CH-evoked FKBP12.6/RyR2 dissociation in PASMCs (Fig. 4f). FK506 at a high dose (1 mg/kg/d) produced a promotable effect on PA remodeling in CH mice, while it at a low dose (0.05 mg/kg/d) exerted a protective effect (Fig. 4g). We further found that this drug at a high and low dose, respectively, enhanced and inhibited CH-induced PH. As shown in Fig. 4h, the mean RVSP was increased from 35.6 ± 3.5 to 43.8 ± 4.6 mmHg at high dose and decreased to 25.7 ± 1.8 mmHg at low dose, respectively ($P < 0.05$); moreover, the similar effects of FK506 on RV weight were observed.

**S107 inhibits PH by blocking RyR2-mediated $Ca^{2+}$ release.** S107, a specific and oral **1,4**-benzothiazepine derivative shown to stabilize FKBP12.6/RyR2 complex, can reduce RyR2-mediated SR $Ca^{2+}$ leak and thereby prevent heart failure progression[13].

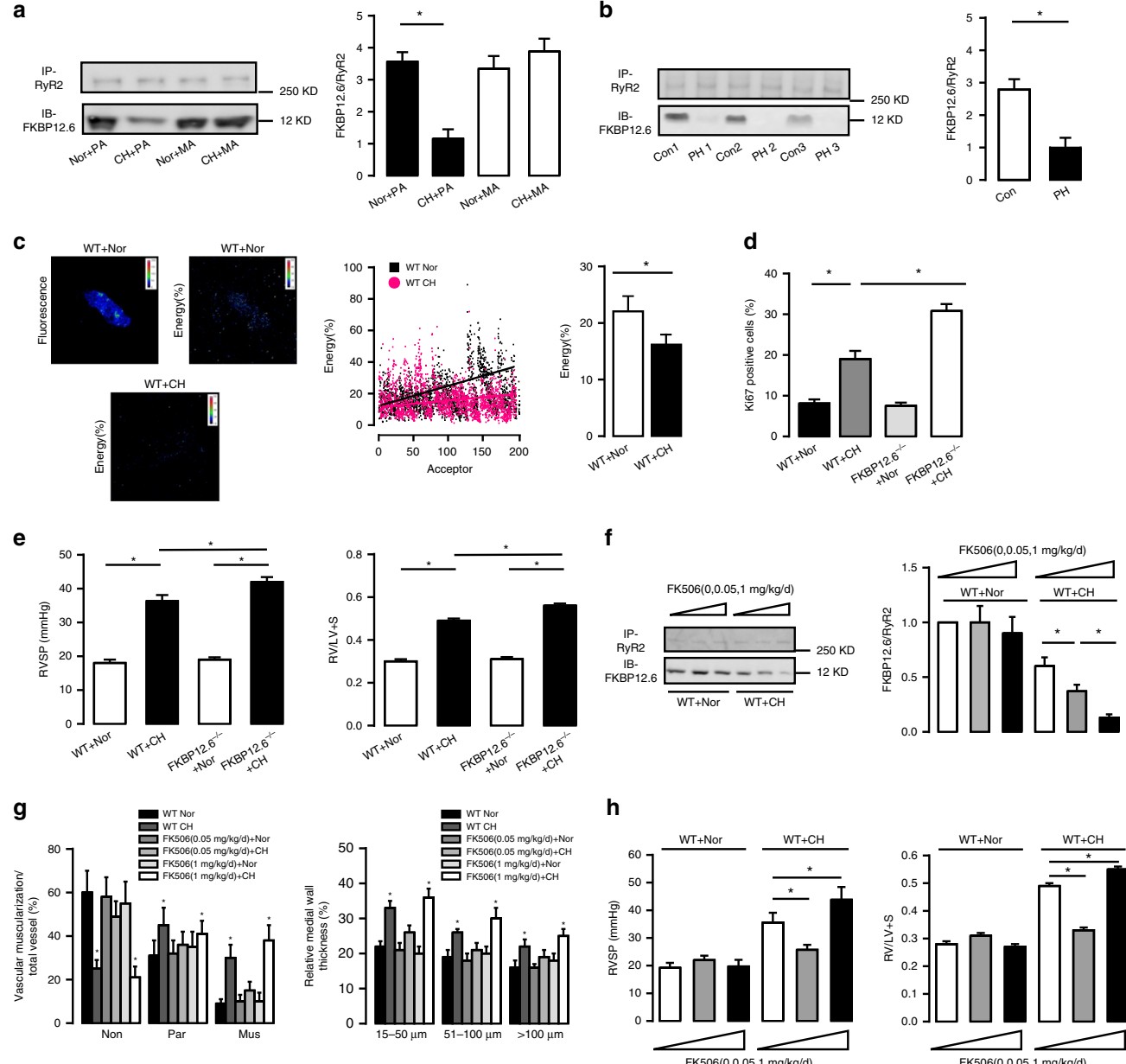

**Fig. 4 CH induces dissociation of FKBP12.6 from RyR2 and the effect of PH. a** Representative Co-IP and western blot of RyR2 and FKBP12.6 from CH induced PH. FKBP12.6 is immunoprecipitated with an anti-RyR2 antibody and then immunoblotted with anti-FKBP12.6 antibody. Tissue is taken from Nor and CH mice's pulmonary arteries (PA) and mesentery arteries (MA) ($n = 3$ independent studies, 5 mice per study). Bar graph summarizes the effect of CH on the association ratio of FKBP12.6 to RyR2. **b** Representative Co-IP and western blot of RyR2/FKBP12.6 complex from patients with PH and age-matched non-PH (Con) human PAs. Bar graph depicts the association ratio of FKBP12.6 to RyR2 ($n = 3$ independent studies). **c** Representative pictures of FRET measurement between donor (FKBP12.6) and acceptor (RyR2) in PASMC. At different levels of acceptor, WT Nor PASMC (black spot) transferred more energy than CH PASMC (red spot), which indicates the experiment is independent of acceptor. Bar graph summarizes average percentage of FRET efficiencies between RyR2/FKBP12.6 pairs in Nor and CH PASMC ($n = 5$ independent studies). **d** In vivo PASMC proliferation (Ki-67 staining) after 3 weeks CH in FKBP12.6$^{-/-}$ mice ($n = 5$ independent studies, 50 vessels per study). **e** Summarized the effect of CH on the RVSP in WT and FKBP12.6$^{-/-}$ PH mice model. Ratio of RV/LV + S to measure RV hypertrophy ($n = 7$ independent studies, 5 mice per study). **f** Representative Co-IP and western blot of RyR2/FKBP12.6 association from low dosage and high dosage FK506 treatment groups. Bar graphs describe the summarized results of FKBP12.6/RyR2 binding ratio in FK506 treating PH model ($n = 4$ independent studies, 5 mice per studies). **g** In vivo pulmonary vascular remodeling evaluated by proportion of non-(Non), partially (Par), or fully (Mus) muscularized PAs and medial wall thickness ($n = 7$ independent studies, 60 vessels per study). **h** Impact of 3-weeks low dosage FK506 (0.05 mg/kg/d) and high dosage FK506 (1 mg/kg/d) on RVSAP and RV/LV + S, indicating low dosage FK506 has protective effect however high dosage FK506 facilitates the development of PH ($n = 7$ independent studies, 5 mice per study). Data are expressed as mean ± standard error. (*$P < 0.05$, using one-way ANOVA test or Student's $t$ test).

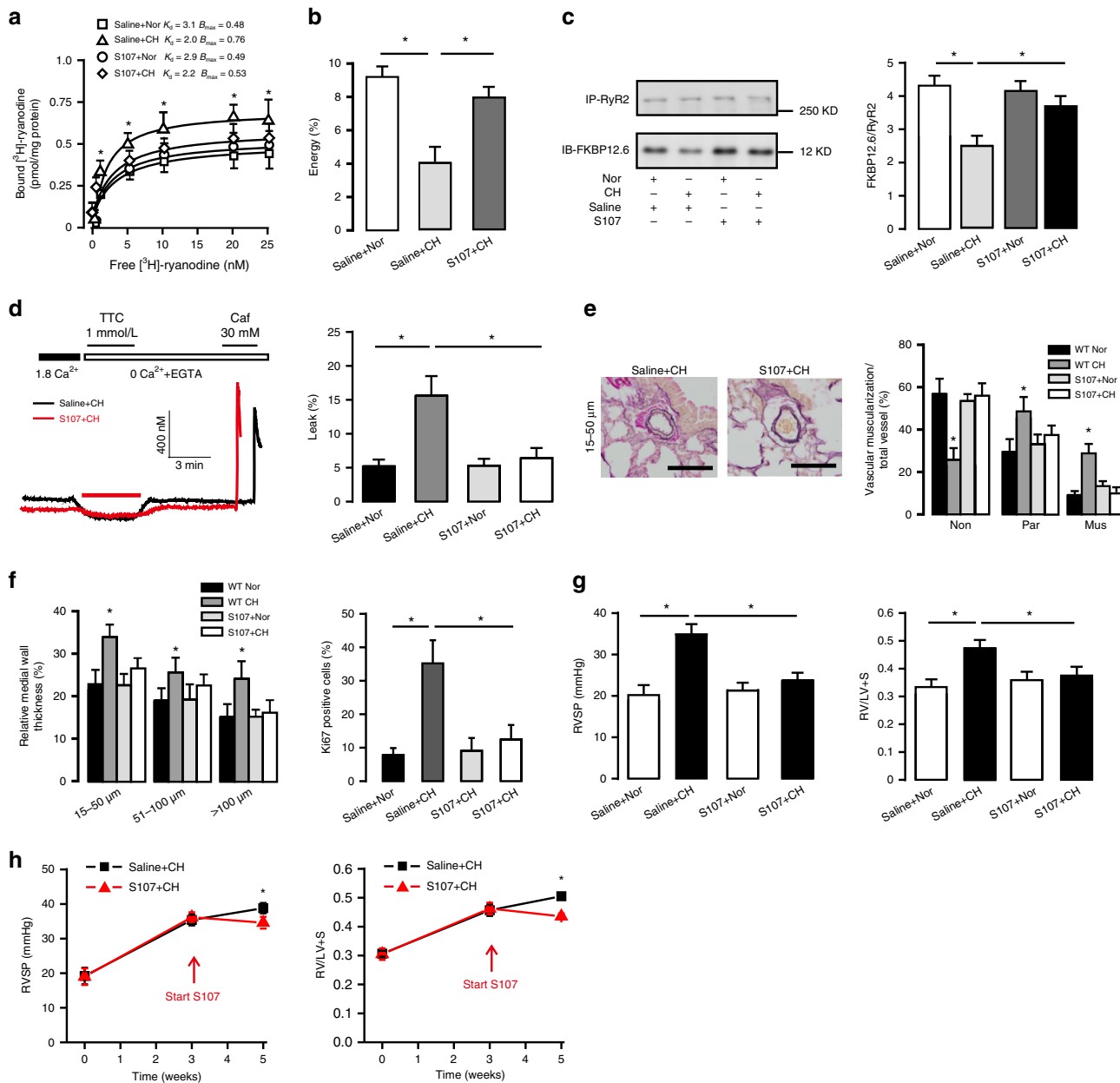

**Fig. 5 S107 prevents the development of PH as well as hyperfunctional Ca²⁺ signaling. a** Determination of RyRs activity by [³H]-ryanodine binding assays in the S107 treating groups ($n = 5$ independent studies, 6 mice per study). **b** The percentage of FRET efficiencies between RyR2/FKBP12.6 pairs ($n = 3$ independent studies, 70 cells per study). **c** Representative immunoblots of RyR2 and FKBP12.6 in S107 treating group showed reduced depletion of FKBP12.6 from RyR2. Bar graph summarizes the association ratio of FKBP12.6 to RyR2 ($n = 4$ independent studies, 5 mice per study). **d** Representative $[Ca^{2+}]_i$ tracing from PASMC loaded by fura-2/AM switch to 0 $Ca^{2+}$ PSS and 1 mmol/L TTC to block RyR2. SR $Ca^{2+}$ content is measured by adding 30 mM caffeine. SR $Ca^{2+}$ leak ratio is quantified and normalized to SR $Ca^{2+}$ storage ($n = 5$ independent studies, 90 cells per study). **e** Typical lung sections depicting pulmonary arteries in S107 treatment group (bar scale: 50 μm). Proportion of non-(Non), partially (Par), or fully (Mus) muscularized PAs in total counted PAs ($n = 7$ independent studies, 60 vessels per study). **f** Medial wall thickness of PAs sized 15–50, 51–100, and over 100 μm in relation to cross-sectional diameter. PASMC proliferation (Ki-67 staining) in vivo after CH ($n = 7$ independent studies, 60 vessels per study). **g** 3 weeks treatment of S107 (20 mg/kg/d) prevents the elevation of RVSP in hypoxia induced PH and RV hypertrophy ($n = 7$ independent studies, 5 mice per study). **h** S107 treatment can prevent the further progression of RVSP and right ventricular hypertrophy (RV/LV + S) ($n = 5$ independent studies, 5 mice per study). Data are expressed as mean ± standard error. (*$P < 0.05$, using one-way ANOVA test).

Herein, we found that in vivo treatment of S107 blocked the increased RyR activity in PASMCs from CH mice (Fig. 5a). In support, S107 treatment completely inhibited CH-caused decrease in FKBP12.6/RyR2 FRET signal (Fig. 5b) and FKBP12.6/RyR2 ratio (Fig. 5c). The increased resting $[Ca^{2+}]_i$ and shortened length (contraction) in PASMCs as well as increased PA contractility in CH mice were all abolished (Supplementary

Fig. 5a–c). As shown in Fig. 5d, the increased ratio of SR $Ca^{2+}$ leak from SR $Ca^{2+}$ storage in PASMCs from CH mice was also fully inhibited by S107 treatment (6.4 ± 1.5 versus 15.6 ± 2.9%, $P < 0.05$). Likewise, CH-evoked muscularization, MWT and SMC proliferation of PAs (i.e., PA remodeling) were all prevented in CH mice after treatment with S107 (Fig. 5e, f). In addition, treatment of S107 in vivo fully abolished the elevated RVSP and

increased RV hypertrophy in CH mice compared with control mice. The mean RVSP was $36.0 \pm 2.5$ and $24.1 \pm 1.5$ mmHg, respectively, and RV/LV + S was $0.38 \pm 0.01$ and $0.48 \pm 0.03$ ($P <$ 0.05, Fig. 5g). Together, S107, like RyR2 KO, restores the aberrant CH-evoked SR $Ca^{2+}$ leak in PASMCs and blocks PA vasoconstriction, remolding, and hypertension.

In established PH mice following CH for 3 weeks, in vivo administration of S107 for 2 weeks in the continued presence of CH lowered RVSP by ~4.5 mmHg and RV hypertrophy ($P < 0.05$, Fig. 5h). S107 treatment also corrected the altered ESPVR and other RV dysfunction indexes (Supplementary Table 2 and Supplementary Fig. 5D). Consequently, S107 shows a potential of its therapeutic effect in the clinical treatment of PH.

**RISP knockdown block CH-induced RyR2 oxidation and PH.** RISP is a primary mediator of hypoxic ROS generation, and RyR2 oxidation is a major mechanism for RyR2 hyperfunctions[8,13,15,28]. We unveiled that intravenous injection of lentiviral RISP shRNAs successfully silenced RISP expression in PASM tissues, but not in liver tissues (Fig. 6a). Importantly, RISP KD suppressed hyperfunctional RyR activity in PASMCs from CH mice (Fig. 6b). ROS-induced proteins oxidation results in specific carbonyl introduction into protein side chains; thus, the determination of protein-bound carbonyls using the highly sensitive 2,4-dinitrophenyl (DNP)-modification is the most commonly used procedure to assess protein oxidation[15]. To determine RyR2 oxidation, RyR2 protein bound by carbonyls was immunoprecipitated, followed by immunoblotting using anti-DNP antibody. As shown in Fig. 6c, significant RyR2 oxidation was detected in PASMCs from CH mice. The RyR2 oxidation in PASMCs was blocked by RISP KD in vivo (Fig. 6d). Moreover, the mean RVSP was reduced from $35.5 \pm 2.5$ mmHg in CH mice

received non-silencing shRNAs to $24.5 \pm 5.0$ mmHg in CH mice received RISP shRNAs ($P < 0.05$, Fig. 6e). In addition to RVSP, RISP KD in vivo also blocked the RV hypertrophy. In agreement with RISP-mediated ROS generation, its KD in vivo almost completely inhibited ROS production in mitochondria and complex III from PASMCs of CH mice (Fig. 6f). This result further indicates the specific role of RISP-dependent ROS in the development of CH-induced PH.

**NF-κB/cyclinD1 pathway mediate RyR2-associated CH-caused PA remodeling.** NF-κB is an important transcriptional factor and regulates cell proliferation and vascular remodeling; upon activation, its necessary subunit p65 and p50 translocate from the cytoplasm to nucleus and then initiate downstream transcription processes[29]. As shown in Fig. 7a, b, cytoplasmic p65/p50 protein expression levels were unchanged in PASMCs from CH mice; however, nuclear p65/p50 expression levels were increased by almost sixfolds (Fig. 7c). CH-caused increases in nuclear p65/p50 expression levels were blocked in PASMCs from $RyR2^{-/-}$ mice (Fig. 7c). Treatment with S107 in vivo produced a similar effect (Fig. 7d). Consistent with the view that IκBα degradation is critical for the translocation of NF-κB, its expression level was largely decreased in PASMCs from CH mice; the CH-mediated reduction in IκBα expression level was blocked in $RyR2^{-/-}$ and S107-treated mice (Fig. 7e). NF-κB-dependent cyclin D1 up-regulation is an important signaling process in the regulation of proliferation of SMCs[26,30]. In conformity, cyclin D1 expression level was remarkably upregulated in PASMCs from CH mice, but not in $RyR2^{-/-}$ and S107-treated mice (Fig. 7f).

Pyrrolidine dithiocarbamate (PDTC), a low molecular weight thiol compound, is known to inhibit NF-κB activation. As shown in Fig. 7g, application of PDTC in vivo significantly suppressed

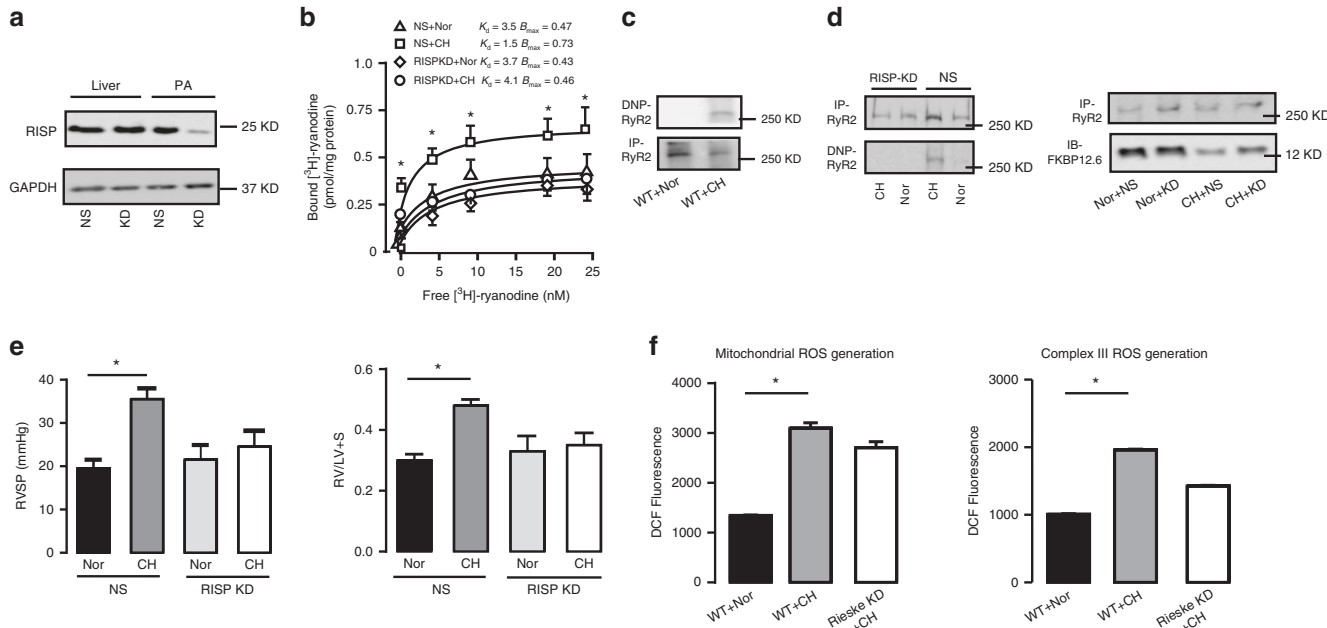

**Fig. 6 Silencing RISP mitigates the hyperfunctional RyR2 activity after CH. a** Representative western blot of in vivo specific silencing RISP in PAs rather than liver using shRNA-mediated lentivirus hydrodynamic intravascular (jugular vein) delivery technique. (NS, shRNA of non-silencing; KD, knock down group). **b** RyR activity in RISP KD group is measured by [3H]-ryanodine binding assays ($n = 5$ independent studies, 5 mice per study). **c** Representative western blot of oxidized RyR2 from PAs after exposure to CH in the same film. The oxidation levels are assayed using an anti-2,4-dinitrophenyl (DNP) antibody against ROS-mediated, DNP-derivatized protein carbonyls. **d** RISP KD suppresses the oxidation of RyR2 and reverses remodeling of FKBP12.6/RyR2 complex determined by western blot. **e** Silencing RISP in vivo prevents the development of CH-induced PH by measuring RVSP and RV/LV + S ($n = 5$ independent studies, 5 mice per study). **f** Silencing RISP inhibits the mitochondrial-mediated and complex III-mediated reactive oxygen species (ROS) generation respectively. Data are expressed as mean ± standard error. (*$P < 0.05$, using one-way ANOVA test).

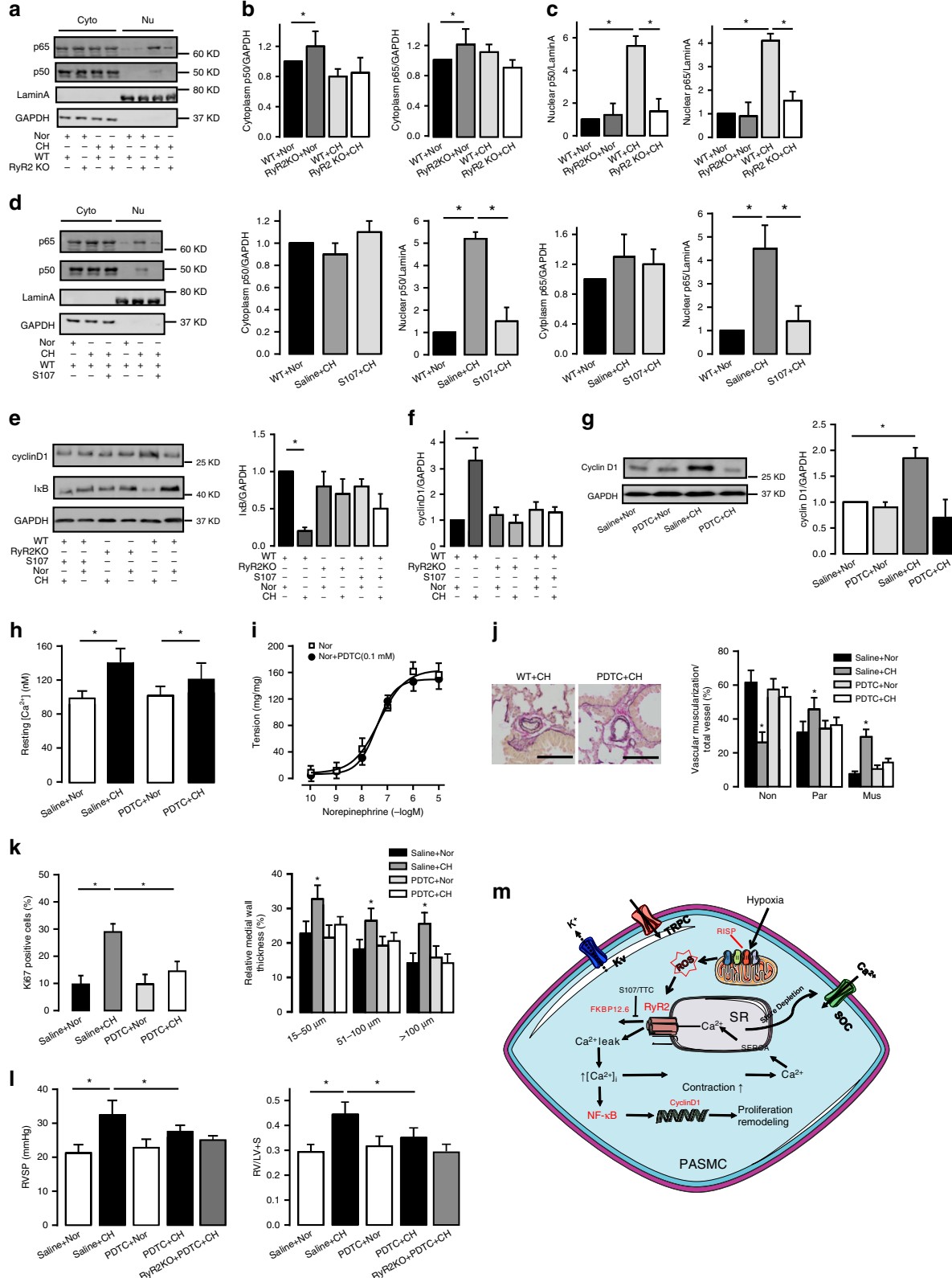

the up-regulation of cyclin D1 in PASMCs from mice with PH. PDTC failed to affect the resting $[Ca^{2+}]_i$ in PASMCs from either control or CH mice (Fig. 7h), but blocked the increased contractility (Fig. 7i). Administration of PDTC (100 mg/kg/d) in vivo prevented PA remodeling in CH mice (Fig. 7j, k). Furthermore, PDTC blocked the elevated RVSP in CH mice compared with control animals (28.0 ± 3.0 versus 34.0 ± 4.0

mmHg, $P < 0.05$), but had no effect in RyR2$^{-/-}$ mice (25.3 ± 3.0 mmHg). In agreement with the effect on RVSP, PDTC produced a similar effect on RV weights in control, CH and RyR2$^{-/-}$ mice (Fig. 7l). Collectively, these results prove that NF-κB functions as a downstream signaling molecule to mediate the essential role of FKBP12.6/RyR2 complex in the development of CH-induced PH (Fig. 7m).

**Fig. 7 CH induces activation of NF-κB (p65/p50)/cyclinD1 pathway in PASMC. a** A representative western blot of p65/p50 translocation from cytoplasm (Cyto) to nuclei (Nu) in RyR2KO mice. **b** Bar graphs depict the expression of p65/p50 in cytoplasm normalized by GAPDH ($n = 3$ independent studies, 5 mice per study). **c** Summarized data of the expression of p65/p50 in nuclei normalized by Lamin A ($n = 3$ independent studies, 5 mice per study). **d** A representative western blot of p65/p50 translocation from cytoplasm (Cyto) to nuclei (Nu) in S107 treating group. Bar graphs depict the expression of p65/p50 in cytoplasm and nuclei in S107 treatment group. The protein levels are normalized by GAPDH and Lamin A respectively ($n = 3$ independent studies, 5 mice per study). **e** A representative western blot of cyclin D1 and IκB expression in RyR2KO and S107 treatment groups. **e** IκB and **f** cyclin D1 expression normalized by GAPDH ($n = 3$ independent studies, 5 mice per study). **g** Western blot shows that PDTC inhibits the over-activation of cyclin D1 in PH model ($n = 3$ independent studies, 5 mice per study). **h** PASMC resting $[Ca^{2+}]_i$ in either Nor or CH is not changed in PDTC (100 mg/kg/d) treatment group ($n = 5$ independent studies, 80 cells per study). **i** Pre-incubation with PDTC (0.1 mM) does not change the contractility of PAs ($n = 6$ independent studies, 5 PA stripes per study). **j** Typical lung sections of pulmonary arteries (bar scale: 50 μm). In vivo evaluation of pulmonary vascular remodeling by measuring proportion of non-(Non), partially (Par), or fully (Mus) muscularized PAs ($n = 7$ independent studies, 60 vessels per study). **k** PASMC proliferation (Ki-67 staining) in vivo in PH models and medial wall thickness of PAs. ($n = 7$ independent studies, 60 vessels per study). **l** Application of PDTC suppresses the elevation of RVSP and RV hypertrophy in PH model ($n = 5$ independent studies, 5 mice per study) without major effect of RyR2 KO mice. **m** Schematic illustration summarizes the proposed model of RISP/ROS-RyR2/FKBP12.6-NF-κB/cyclin D1 pathway. Data are expressed as mean ± standard error. (*$P < 0.05$, using one-way ANOVA test).

## Discussion

$Ca^{2+}$ signaling in PASMCs plays an essential role in the regulation of PA vasoconstriction and remodeling that lead to the development of PH; this well-recognized $Ca^{2+}$ signaling is precisely generated and regulated by multiple ion channels in the cell. However, it remains unclear which ion channel is most important in the development of PH. Our current study has identified that RyR2 plays an important role in mediating PH. The first evidence in support of this concept is that the activity of RyRs, determined using [³H]-ryanodine binding assay, is significantly increased in PASMCs from mice with CH-induced PH. Our previous studies and others have demonstrated that all three RyR subtypes (RyR1, RyR2, and RyR3) are expressed in PASMCs; among them, RyR2 is the most dominantly expressed[5,6], similar to the result in airway SMCs[31–33]. A series of our previous in-vitro studies further indicate that all three RyR subtypes are involved in hypoxia-induced $Ca^{2+}$ and contractile responses in PASMCs, but RyR2 is a very important player[7–9]. Indeed, herein we have found that the increased RyR activity and associated $Ca^{2+}$ signaling are blocked in PASMCs from SMC-specific RyR2⁻/⁻ mice. We have reported that the caffeine-mediated $Ca^{2+}$ transients are not affected by IP₃ in PASMCs[9]. Moreover, our previous reports reveal that all three IP₃R subtypes (IP₃R1, IP₃R2, and IP₃R3) are expressed in PASMCs[11,34]; however, only IP₃R1 may play an important role in hypoxic $Ca^{2+}$ and contractile responses[34]. Apparently, further experiments are needed to disclose the functional role of IP₃R1 in the development of hypoxic PH.

In line with the increased RyR2 activity and function, the baseline tension (length) and $[Ca^{2+}]_i$ are higher in PASMCs from PH mice relative control animals. Our data further reveal that agonist-induced increase $[Ca^{2+}]_i$ is enhanced in PASMCS from mice with PH. All these cellular responses are eliminated in PASMCs from RyR2⁻/⁻ mice, further supporting that RyR2 in PASMCs is hyperfunctional in PH. Interestingly, the SR $Ca^{2+}$ content is substantially decreased in PH mouse PASMCs. RyR2⁻/⁻ mice are resistant to CH-induced PH. In support, RyR2 has been reported as a very important player in the hypoxic increase in $[Ca^{2+}]_i$ in PASMCs[5,6,35,36].

Our experiment further reveals that RyRs become hyperfunctional in the model of CH-induced PH. However, the subtype of RyRs that participates in the development of CH-induced pulmonary arterial remodeling is still unknown. Accordingly, we have created SMC-specific RyR2⁻/⁻ mice. SMC-specific RyR2⁻/⁻ mice produce no overt phenotype in normoxic condition, but protective effects on CH and SU5416/CH, which may both be mediated by inhibition of SR $Ca^{2+}$ release. This results suggest that the over-activation of RyR2 and increased $[Ca^{2+}]_i$ are the key

processes in the development of PH, like other cardiovascular diseases such as heart failure[13] and cerebral arterial disease[37]. It is known that how the expression of RyR2 changed under pathophysiological condition is controversial[37,38]. We have found that the increased RyR2 activity is due to the remodeling of its regulators (e.g., FKBP12.6) rather than the alteration of expression level. Kaßmann et al.[39] have reported the successful generation of tamoxifen-inducible SMC-specific RyR2-deficient mice. In their report, the acute hypoxia-induced increase in PA pressure in isolated perfused lungs is enhanced in RyR2⁻/⁻ mice. Intriguingly, we have found that a significant reduction of acute hypoxia-induced increase in PA pressure in vivo in RyR2⁻/⁻ mice (Supplementary Fig. 6a–c). It should be aware that tamoxifen itself influences the development of PH[40] and the in vivo model is used in our study. These facts may well explain the difference between the report by Kaßmann et al. and ours.

To determine the mechanism by which the RyR2KO mice manifest the resistance in the hypoxic condition, we tested the vascular contractility, resting $Ca^{2+}$ level, SR leak, SR $Ca^{2+}$ release, and RyR activity. The increased SR leak and depleted $Ca^{2+}$ storage in PASMCs had been observed in established PH. TTC, which is used in clinical practice and relatively specific RyR2 inhibitor, shows a significant protective effect in PH. Our current report validates that RyR2-associated $Ca^{2+}$ release could be the upstream signal to regulate NF-κB activation and subsequently affect downstream $K_V$ and TRPC channel expression and activity; thus, RyR2 serves as a primary molecule in the hypoxic calcium signaling alternations in PASMCs.

FKBP12.6 has a high affinity with RyR2, playing as a stabilizer of RyR2. However, there is debating about the role of FKBP12.6 in disease development. A series of studies by Marks and colleagues[13,41–43] have shown that FKBP12.6 mediates cardiac arrhythmia, heart failure, and sudden cardiac death, which are attributed to dysfunction of the mechanism of RyR2 inhibition by FKBP12.6. On the other hand, Bers and associates[44] have found that only a small portion of endogenous RyR2 is associated with FKBP12.6 in cardiac myocytes, and the physiological level of FKBP12.6 might not affect RyR2 activity. Other reports have questioned the role of FKBP12.6 in cardiac cells[17–20]. Collectively, further investigations are warranted for clarification of the current controversy on the RyR2/FKBP12.6 interaction in specific cell types.

A study using FKBP12.6 KO mice has shown that FKBP12.6 is associated with glucose metabolism by its interaction with RyR2 in pancreatic islets[45]. We have previously reported that FKBP12.6, but not FKBP12 protein, is highly expressed and associated with RyR2, thereby playing a significant role in PASMCs[11]. Similar findings have been made in airway SMCs by

us[12] and other investigators[46]. In this study, we demonstrate that FKBP12.6$^{-/-}$ mice have the increased sensitivity to CH-induced PH, although these mice have no effect under normoxic conditions. FKBP12.6 is the intracellular target for FK506. As an immunosuppressant drug widely used in transplantation, FK506 produces side effects including hypertension and diabetes by causing Ca$^{2+}$ signaling dysfunction in SMCs[47,48]. We have found that application of FK506, which binds to FKBPs and dissociates these proteins from RyR, enhances acute-hypoxia-evoked Ca$^{2+}$ release in isolated PASMCs[8,11]. However, the effect of chronic administration of FK506 on PH in vivo is uncertain. Dr. Rabinovitch's group[49] have reported that administration of FK506 at low dosage (0.05 mg/kg/d) shows a protective function in PH possibly by activating BMPR2 signal. A similar result has been observed in this study. Besides, we have found that that FK506 at a high dose (1 mg/kg/d) promotes the development of hypoxic PH. Thus, FK506 produces a biphasic effect on PH. Low dose FK506 is currently used in clinical trials and has shown a promising outcome[50,51]. In addition, we have not observed any phenotype of FKBP12.6$^{-/-}$ mouse or effect of FK506 treatment under normoxic condition. Apparently, FKBP12.6 removal cannot induce PH on its own. Accordingly, both FKBP12.6 and RyR2 contribute to the development of hypoxic PH. S107 is known to specifically stabilize FKBPs/RyR complex and inhibit SR Ca$^{2+}$ leak. Multiple publications indicate that S107 inhibits the development and progression of multiple diseases[13,16]. Our study reports that S107 prevents the development of PH through stabilization of FKBP12.6/RyR2 complex. As a benzothiazepine derivative, S107 may display pleiotropic effects, but the findings from the current study unveil that this small molecule at least provides a potential therapeutic target in treating PH. Collectively, our previous investigations and current studies provide strong evidence that inhibition of RyR2 by FKBP12.6 underlies the pathological responses in PAMCS as diagrammed in Fig. 7m. These findings suggest that this mechanism may be tissue- and/or condition-specific.

Although how Ca$^{2+}$ signaling becomes hyper-functional in PASMCs are still uncertain, increasing evidence indicates that intracellular ROS play a critical role in mediating the development and progress of PH by regulating multiple ion channels[2,3,22]. We[21,28] and others[23] demonstrate that a unique protein in the mitochondrial complex III, namely RISP, serves as an essential molecule in the hypoxic mitochondrial and intracellular ROS generation in PASMCs. In this study, we have further identified that RISP is a key upstream player to FKBP12.6/RyR2 complex and thus has an essential role in CH-induced PH by mediating ROS generation. Indeed, lentiviral shRNA-mediated RISP silencing interrupts the positive feedback between ROS production and Ca$^{2+}$ release but fails to cause overt deficiency and mitochondrial dysfunction in vivo. It provides a therapeutic drug target in treating PH without a great potential side effect.

Regardless of different genetic or pathogenic factors, the pathogenesis of PH can be attributed to increased pulmonary arterial contractility and uncontrolled PASMC proliferation[52]. Cell proliferation is regulated by multiple transcription factors, among which NF-κB may play a significant role in PH[24,25,53] and its specific inhibitor PTDC blocks hypoxia-induced PH in rats[54]. However, whether dysfunctional Ca$^{2+}$ signal transmit its effect through NF-κB is uncertain. Our previous report indicates the link between TRPC channel and NF-κB in the development of asthma[55]. Herein, we provide evidence showing that NF-κB and cyclin D1 activation are important for CH-induced PA remodeling and hypertension. These results are consistent with their overall functions in the increased proliferation of PASMCs and associated PH[26,30,56]. Administration of PDTC, a potent inhibitor of NF-κB, fails to affect the upstream Ca$^{2+}$ signal. RyR2 KO plus PDTC treatment, similar to RyR2 KO or PDTC alone, abolishes hypoxic PH in mice. All these findings further support our concept that NF-κB is a downstream molecular target of RyR2 signaling. It may also need to point out that NF-κB plays diverse and important functions in vivo; thus, it is not an ideal drug target for PH. Molecules modulated by NF-κB and those with more specific expression in PASMCs will be our focus of future study.

In summary, we have demonstrated the functional importance of FKBP12.6/RyR2 complex in the development of PH in both animals and human samples, in which CH dissociates FKBP12.6/RyR2 complex, causes SR Ca$^{2+}$ leak and increases [Ca$^{2+}$]$_i$ in PASMCs, thereby leading to subsequent PA remodeling and vasoconstriction. These events may occur due to the mitochondrial RISP-dependent ROS-mediated RyR2 oxidation. Downstream activation of NF-κB/cyclin D1 signaling as a result of the oxidation-evoked RyR2 hyperfunction and Ca$^{2+}$ singling is directly involved in PASMC proliferation and PA remodeling. The major strength of our study also lies in the evidence that genetic and pharmacological inhibition of RyR2, stabilization of FKBP12.6/RyR2 complex by treatment of S107, and in vivo lentiviral shRNA-mediated knockdown of RISP all can block the development of PH, providing effective strategies to prevent and treat PH in humans.

## Methods

**Materials and animal.** Collagenase, dithiothreitol, dithioerythritol, myxothiazol, papain, TTC, PDTC, tacrolimus (FK506), and SU5416 (SUGEN; S8442) were purchased from Sigma-Aldrich Corp; anti-FKBP12/12.6 and anti-RyR2 antibodies from ABR Affinity Bio-Reagents Products; p50, α-actin, cyclin D1, IκBα, and vWF, and GAPDH antibodies from Santa Cruz Biotech, Inc; p65 antibody from Cell signaling Technology, Inc; Lamin A and anti-Ki67 antibodies from Abcam; Kv1.5, TRPC1, and TRPC6 antibodies from Alomone Labs; fura-2/AM from Molecular Probes; [$^3$H]-ryanodine from PerkinElmer Corp; S107 (**2,3,4,5**-tetrahydro-**7**-methoxy-**4**-methyl-**1,4**-benzothiazeping) from Cayman Chemical. All procedures were approved by the Institutional Animal Care and Use Committee of Albany Medical College. *C57BL/6*, FKBP12.6$^{-/-}$, SMC-specific RyR2$^{-/-}$ and their corresponding control (WT) mice with the same background and age were used. The generation and maintenance of FKBP12.6$^{-/-}$ mice were reported in our previous publications[11,57]. SMC-specific RyR2$^{-/-}$ mice were produced by breeding *ryr2*-floxed mice with *myh11-cre* mice. The *ryr2*-floxed mice were created by inserting a *lox*P sequence into the 5′-untranslation region exon 1, the first intron of mouse *ryr2*[58,59]; the production of *myh11-cre* mice was reported[60]. Forward and reverse primer sequences for genotyping mice were, respectively, 5′-CCAATTTACTGA CCGTACACC-3′ and 5′-GTTTCACTA TCCAGGTTACGG-3′ for Cre, 5′-GCA CCCTGGGGGCAGCCTTCTCAGC-3′ and 5′-ACATGTGTGCAGGTGTGCGG GTCTG-3′ for WT, 5′-TCCTCGTGCTTTACGGTATCGCCG-3′ and 5′-GCAC CCTGGGGGCAGCCTTCTCAGC-3′ for RyR2 heterozygote, and 5′-GAACAG TTCCTCGCCCTTGCTCAT-3′ and 5′-GAGCCCCTAGAACATCCTGGTTAGC-3′ for RyR2 homozygote. Human PA specimens were obtained from the National Disease Research Interchange (NDIR) or Albany Medical Center Hospital and used according to the protocols approved by the Institutional Review Board for the Protection of Human Subjects in Research at Albany Medical College. PH was diagnosed by physical exam, medical history review and various tests. The PA specimens from age-matched subject who did not have any detectable cardiovascular diseases were used control. Written informed consents were obtained from all human subjects.

**CH-induced PH model.** RyR2 KO and WT male mice of 10–12-weeks old were placed in norm baric chambers for 21 days. The oxygen concentration in the chamber was maintained at 10% (Biospherix Ltd.). Some groups of mice were injected subcutaneously with a single weekly SU5416 (20 mg/kg) combined with 3 weeks of CH[61]. Mice were anesthetized with an intraperitoneal (i.p.) injection of avertin (250 mg/kg). The chest was opened, and a 1.2F pressure–volume catheter (Scisense, Canada) was introduced for measuring RVSP and heart function[62]. Right ventricular (RV) hypertrophy was measured by weighting the ratio of RV to left ventricle (LV) with septum (S) (RV/LV + S). The lung tissue was fixed in 4% paraformaldehyde and process into 5-μm paraffin sections. Sections were stained with Elastic Van Gieson methods[63]. Medial wall thickness (MWT) of pulmonary arteries with different cut angles and crinkling were computed according to the methods of Barth et al.[64] using ImageJ. The proliferation studies, lung sections were deparaffinized in xylene, rehydrated and blocked endogenous peroxidase with 3% H$_2$O$_2$. Slides were incubated overnight with anti-Ki67 rabbit antibody (1:500,

Abcam Inc.). The signal was then developed with immunoperoxidase reagent (ABC-HRP, Vector Labs) and DAB (Sigma-Aldrich) as the substrate.

**S107, PDTC, TTC, and FK506 treatment**. S107 (20 mg/kg/d)[13], TTC (2 mg/kg/d) and PDTC (100 mg/kg/d)[53] were administered using osmotic pumps (Alzet Model 1004) dissolved by 0.9% saline before putting into hypoxic chamber for 3 weeks. The control groups were filled with 0.9% saline. They were implanted subcutaneously on the dorsal surface using a horizontal incision at the neck according to the manufacturer's instruction. Low dosage FK506 (0.05 mg/kg/d) or high dosage FK506 (1 mg/kg/d) were dissolved in 0.9% saline and delivered by daily i.p. injection for 3 weeks during in hypoxic chamber[47].

**Preparation of isolated pulmonary artery (PA) tissues and smooth muscle cells (PASMCs)**. PA SM tissues and cells were prepared from mice[8]. Briefly, mice were euthanized by intraperitoneal injection of sodium pentobarbital. Resistance PAs less than 200 μm were isolated and placed in cold physiological saline solution (PSS) containing (in mM) 125 NaCl, 5 KCl, 1MgSO$_4$, 10 HEPES, 1.8 CaCl$_2$, and 10 glucose (pH 7.4). Then arteries were digested using two-step enzymatic method. Enzyme were dissolved in low Ca$^{2+}$(100 μM) PSS containing (mg/ml) 1.5 papain, 0.4 dithioerythritol, and 1.0 bovine serum albumin (BSA) for 15 min, followed by 1.0 collagenase II, 1.0 collagenase F, 1.0 dithiothreitol, and 1.0 BSA for 15 min. The digested PAs were gently triturated to harvest single SMCs. Dissected PAs were homogenized in buffer containing 0.29 M sucrose and 3 mM imidazole (pH 7.4) for purify SR membrane[8]. After adding protease inhibitor mix and grinding, the homogenate from PAs was centrifuged at $5000 \times g$ for 10 min at 4 °C. Then the supernatant was ultra-centrifuged at $100,000 \times g$ for 45 min at 4 °C to collect the supernatant as the cytosol fraction and the pellet as the SR membrane fraction.

**Muscle tension measurement**. Muscle tension in isolated PA rings were measured[9,21]. PA rings 3 mm in length were placed in 2 ml tissue bath (Radnoti) at 37 °C. The endothelium was removed and confirmed by the absence of relaxation in 10 μM methacholine. Passive tension of 600 mg was applied to each of the PA rings. The viability of the PAs was tested with 10 μM phenylephrine. Experiments started after an equilibration period for 90 min. Contraction recordings were made using highly sensitive force transducer (Harvard Apparatus) with a Powerlab/4SP recording system (AD Instruments). In examine norepinephrine or endothelin-1 (ET-1)-induced muscle contraction, contraction-response curves were constructed by fitting the mean values of data with Origin Version 7 software (OriginLab Corp.) using Boltzmann equation.

**Intracellular Ca$^{2+}$ and SR Ca$^{2+}$ leak measurement**. Measurement of $[Ca^{2+}]_i$ was made using dual excitation wavelength fluorescence method in our lab[32], with TILLvisION digital imaging system (TILL Photonics GmbH). Freshly isolated PASMC with the fluorescent Ca$^{2+}$ indicator dye fura-2/AM (5 μM) at room temperature for 30 min. The fluorescent dye was alternatively excited at 340 nm and 380 nm, and emitted fluorescence was detected at 510 nm. The background signal was corrected by fluorescence recorded in non-cell regions. In calibration experiments, 340 and 380 nm fluorescence at Ca$^{2+}$-free and saturating Ca$^{2+}$ concentrations were determined using nominally Ca$^{2+}$-free PSS containing 10 mM ionomycin and 10 mM EGTA in PSS containing 10 mM ionomycin and 10 nM Ca$^{2+}$, respectively.

SR Ca$^{2+}$ leak in PASMC were measured as following. PASMC were incubated in 1.8 mM Ca$^{2+}$ PSS until steady state $[Ca^{2+}]$i was monitored. Then switching to a 0 Na$^+$, 0 Ca$^{2+}$ with 0.1 mM EGTA in PSS blocked the Ca$^{2+}$ exchange via Na$^+$/Ca$^{2+}$ exchanger and L-type Ca$^{2+}$ channel. Application of TTC (1 mmol/L) was used to antagonize RyR2, followed by administration of caffeine (30 mmol/L) to evaluate SR Ca$^{2+}$ content. The ratio of TTC-dependent resting $[Ca^{2+}]_i$ shift to SR Ca$^{2+}$ content was SR leak[14].

**Fluorescence resonance energy transfer (FRET)**. Fresh isolated PASMC were fixed with 4% paraformaldehyde and permeabilized by 0.5% TritonX-100. The cells were incubated with RyR2 antibody (1:100, mouse monoclonal, Thermo Scientific) and FKBP12.6 (1:100, rabbit monoclonal, Thermo Scientific) together overnight. Then RyR2 will act as donor using anti-mouse Alexa 488 labeled Whole/F(ab)$_2$ fragments antibody (1:250, Invitrogen), and FKBP12.6 act as acceptor using anti-rabbit Alexa 555 labeled F(ab)$_2$ antibody(1:250, Invitrogen). FRET will be measured using LSM-510 laser scanning confocal microscope (Carl Zeiss) and the result will be analyzed using ImageJ with FRET imaging macro plug-in.

**Co-immunoprecipitation (Co-IP)**. PA homogenates (200 μg) were used to immunoprecipitate RyR2 with anti-RyR2 antibody (Thermo Scientific). Then add the protein-G (Santa Cruz Biotech) and incubate at 4 °C overnight. The immunoprecipitates were washed and then elute by adding Laemmli sample buffer. After the samples were boiled for 5 min, equal homogenate was separated by SDS-PAGE (6% gel for RyR2, 15% gel for FKBP12.6). Proteins were transferred to PVDF membrane and develop using anti-RyR2 antibody (1:1000), anti-FKBP12.6 antibody (1:800), then visualized by the chemiluminescence detection kit[13].

**Protein oxidation test**. Protein oxidation was detected by reaction with 2,4-dinitrophenyl hydrazine (DNP) using an OxyBlot™ Protein Oxidation Detection Kit (Chemicon International, Inc.)[8]. According to the manufacturer's instruction, isolated SR membrane samples (30 μg) were denatured by 5 μl 12% SDS and then derivatize with 10 μl $1 \times$ DNP for 15 min at room temperature. The reaction was stopped by adding 7.5 μl neutralization solution. The proteins were electrophoresed on a SDS-PAGE gel and transfer onto PVDF membrane. The membrane was incubated with an anti-DNP antibody (1:150) for 2 h. Blots were visualized and analyzed. The same membrane was incubated with anti-RyR2 antibody (1:1000) for reblotting.

**Western blot analysis of PASMC cytosol and nuclear fraction**. Dissected intrapulmonary PAs were homogenized in buffer containing 0.29 M sucrose and 3 mM imidazole (pH 7.4)[8]. The cytosolic and nuclear extract were separated using nuclear extraction kit (Signosis, Inc., CA) according to the manufacturer's manual. The routine western blot is performed using P50 antibody (Santa Cruz Biotech, Inc.), p65 antibody (Cell signaling Technology, Inc.). GAPDH (Santa Cruz Biotech, Inc.) and Lamin A (Abcam) were used as cytosolic and nuclear loading marker respectively.

**Lentivirus-mediated gene silence and intrajugular injection**. The pGIPZ lentiviral vectors containing RISP shRNA (sequence: CCCTGAATGGGTTATTCTG ATA) and scrambled shRNAs were purchased from ThermoScientific Open-Biosystems (Mouse V2Lmm-72156) and were packaged using pCMV-dR8.2 dvpr and pCMV-VSV-G vectors[28,55,65]. The concentrated lentiviruses containing shRNAs were diluted to a working titer of $3 \times 10^9$ transducing unit (TU)/ml in PBS. Mice were anesthetized by intraperitoneal injection of avertin (250 mg/kg) and placed in a ventral recumbent position on a heating pad with 37 °C. A small incision was made above the neck to expose a jugular vein. A 300 μl solution containing either lentiviral RISP or scrambled shRNAs at $3 \times 10^8$ TU/ml was administrated to each individual mouse via an intrajugular vein using 30-gauge needle through pectoral muscle. The injection was manually performed within a 2-min period of time. After that, the injection site was gently pressed by cotton tip for 5 min, skin incision closed by suture, and mice were released from restraining and warmed by heating lamp until they were recovered from anesthesia. After injection for 24 h, mice were exposed to CH or normoxia for 3 weeks.

**[$^3$H]-ryanodine binding assay**. Isolated PAs were homogenized and concentrated through gradient centrifugation[8]. Eventually, the enriched SR membrane fraction was obtained by ultra-centrifuge ($100,000 \times g$). Each fraction (10 μg) was incubated with each group of [$^3$H]-ryanodine ranged from 0.5–25 nM for 3 h in a binding assay solution (300 μl). The binding mixture was diluted with ice-cold washing buffer and immediately filtered through Millipore Membrane filters presoaked with washing buffer. The radioactivity associated with the filters was determined by liquid scintillation counting. Nonspecific binding was determined in the presence of 50 μM unlabeled ryanodine. All binding assays were executed in duplicate.

**Detection of ROS production**. Isolated PASMCs, mitochondria (using differential centrifugation) or complex III extract (using immunocapture kits from Mitoscience) with 50 μg protein were added to microplate wells containing respiration buffer. After incubation for 10 min, fluorescence was measured using FlexStation-III spectrophotometer with excitation wavelength of 485 nm and emission wavelength of 532 nm. ROS production was determined by subtracting the fluorescence intensity measured in control wells[21].

**Pulmonary arterial endothelial cells (PAECs) isolation**. The procedure was conducted following the protocol from Seybold et al.[66]. The isolated pulmonary arteries PAs) were washed with PSS (37 °C) and put a piece of thread tightly round one end of PA. Fill the PA with warm collagenase and incubate at 37 °C for 15 min then rinse and massage the artery with 10 ml culture medium. Centrifuge the rinse medium at 300 g for 10 min, discard the supernatant and repeat for three times. Then the PAECs were ready for further studies.

**Acute hypoxia induced change of RVSP**. When the 1.2F pressure–volume catheter (Scisense, Canada) was introduced for measuring RV pressure with room air, the baseline was recorded. The mice were then ventilated with hypoxic gas (5% O$_2$/95% N$_2$) for 1–3 min and then switched back to room air to allow recovery. The change in the RVSP was measured[23].

**Immunofluorescence staining**. Cells were placed on coverslips precoated with fibronectin (40 μg/ml) in phosphate-buffered saline (PBS) for 30 min at room temperature, fixed with 4% paraformaldehyde for 15 min, and permeabilized with 0.2% Triton. After incubated in 2.5% BSA in PBS blocking solution for 30 min, cells incubated with anti-α-actin (1:250, SMC marker), anti-RyR2 antibody (1:150) and anti-vWF antibody (1:250, endothelial cell marker) for overnight at 4 °C, followed by Alexa488- and Alexa594-conjugated antibodies for 90 min. The staining was examined using LSM510 laser scanning confocal microscope (Carl Zeiss). The nucleus was stained by **4,6**-diamidino-**2**-phenylindole (DAPI)[6–9].

**Statistical analysis**. All data were analyzed using Origin 8.5. Data are expressed as mean ± standard error. The mean values were obtained from at least three independent experiments. Student's $t$ test for comparisons before and after treatment in the same sample, unpaired (independent) Student's $t$ test for 2-sample comparisons, one-way ANOVA followed by Tukey's multiple-comparison post hoc analysis for comparisons of the means that were classified in two different ways or the mean responses in an experiment with two factors. Values of $*P < 0.05$ was accepted as statistically significant.

**Reporting summary**. Further information on research design is available in the Nature Research Reporting Summary linked to this article.

## Data availability

The authors declare that all relevant data supporting the finding of this study are available within the paper and its supplementary information files. All data are available from the corresponding author on reasonable request. Source data are provided with this paper.

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

## Acknowledgements
This work was supported by AHA Established Investigator Award 0340160N (Y.-X.W.) and Scientist Development Grant 0630236N (Y.-M.Z.), JSPS Core-to-Core Program (H.T.), and NIH R01HL64043, HL064043-S1, HL108232, and HL122865 (Y.-X.W.)

## Author contributions
Conceptualization, L.M., Y.-M.Z., and Y.-X.W.; Investigation and Validation, L.M., Y.-M.Z., T.S., V.R.Y. LJ., L.T., M.M.B., H.T. and M.A.J.; Writing-Original Draft, L.M., Y.-M.Z., L.T. and Y.-X.W.; Writing-Review & Editing, L.M. Y.-M.Z. S.K. and Y.-X.W.; Funding Acquisition, Y.-M.Z. and Y.-X.W

## Competing interests
The authors declare no competing interests.
