## [Peer Review File · Nature Communications]

Reviewers' Comments:

Reviewer #2:

Remarks to the Author:

In the current manuscript by Mei et al., entitled "Rieske iron-sulfur protein induces remodeling of FKBP12.6/RyR2 complex and subsequent pulmonary hypertension through NF- κ B/cyclin D1 pathway", the authors investigate pathogenic links between Ca²⁺ signaling and its downstream signaling proteins and transcription factors in the development of pulmonary hypertension. A rise in intracellular [Ca²⁺] ([Ca²⁺]_i) in PASMC is a major trigger for pulmonary vasoconstriction and important stimulus for pulmonary vascular remodeling in patients with pulmonary arterial hypertension and animals with experimental pulmonary hypertension. The results from this study demonstrate a very interesting link between the RISP/ROS-mediated interaction of RyR2 and /FKBP12.6 with the Ca²⁺-sensitive NF- κ B/cyclin D1 signaling in PASMC proliferation and pulmonary vascular remodeling. The data presented in the study are novel and significant, and of high quality. The evidence for the novel link is compelling given the combined use of various techniques and in vitro/in vivo models in the study. I only have a few minor concerns that may need authors' attention to improve the manuscript.

Minor comments:

1. Please include molecular-weight size marker in Western blot images (so to allow readers to judge or estimate the size of target bands) shown in Figures 1(D), 4(A-B and F), 5(C), 6(A, C, D), 7(A, D, E, G) and S1 (A).
2. Although the study focuses on the RyR2-mediated Ca²⁺ signaling, it is important to show whether caffeine (or ryanodine)-mediated Ca²⁺ transients are affected by IP3 and whether other RyRs and IP3Rs are expressed in the cells used in the study.
3. It would be informative to discuss whether the RISP/ROS-mediated effect on RyR2/FKBP12.6 interaction (or dissociation) is unique to hypoxia-induced mitochondrial ROS production and hypoxia-induced pulmonary vasoconstriction/vascular remodeling. Cytoplasmic ROS can be increased by many extracellular and intracellular stimuli.

Reviewer #3:

Remarks to the Author:

The manuscript by Mei et al. reports that the Rieske iron-sulfur protein (RISP) induces remodeling of FKBP12.6/RyR2 complex and subsequently pulmonary hypertension (PH) through NF- κ B/cyclin D1 pathway. They elegantly identified that smooth muscle cell (SMC)-specific RyR2 knockout (KO) restores altered Ca²⁺ signaling, inhibits increased nuclear factor (NF)- κ B and downstream cyclin D1 activation as well as cell proliferation, and prevents PA vasoconstriction, remodeling and hypertension in CH-induced PH mice. FKBP12.6 KO or FK506 treatment enhances the sensitivity to CH-induced PH, while treatment of S107 (a specific stabilizer of RyR2/FKBP12.6 complex) produces an opposite effect. Lentiviral shRNA-mediated SMC-specific RISP knockdown can diminish mitochondrial reactive oxygen species (ROS) generation, RyR2/FKBP12.6 complex dissociation, and RyR2 activation, thereby preventing PH in CH mice. The manuscript is very convincing, highly relevant and comprehensive although I believe there are mainly novelty issues that are required to be addressed.

Major:

1. The authors conclude that to their knowledge, this is the first study to genetically and conditionally deplete RyR2 gene in smooth muscle cells in vivo. This is entirely misleading, as there is a study exploring the role of RyR2 in elementary Ca²⁺ Signaling in pulmonary arteries, pulmonary vasoconstriction (HPV) and PH. This is intentionally misleading to support a novel mechanism and not necessary. Targeted deletion of the RyR2 gene resulted in the enhanced pulmonary artery pressure response to sustained hypoxia (See Kaßmann M et al. J Am Heart Assoc. 2019 May 7;8(9):e010090). The authors do show a very novel aspect that could be important in pulmonary vascular remodeling underlying PH.

2. Similarly, existing reports already suggest that RISP is a very important player in hypoxic ROS and Ca²⁺ signaling in PSMCs (See Song et al. Adv Exp Med Biol. 2017;967:289-298). The authors have extended these in vitro findings to in vivo studies.

3. In addition, there is a study which explored the role of RyR2/FKBP12.6 complex and the influence of FK506 in the context of HPV. This study identified that there is neither effect of displacement of FKBP12.6 from RyR2 nor FK506, a drug which displaces FKBP12.6 from ryanodine receptor 2 (RyR2), had any effect on HPV (See Connolly et al. J Physiol. 2013 Sep 15;591(18):4473-98).

4. Similarly, the influence of PTDC (targeting NfKB signaling) on chronic hypoxia induced PH is not novel (see Fan J et al. High Alt Med Biol. 2016 Mar;17(1):43-9).

Based all the above-mentioned points and published literature in the field of RyR2 and HPV, the novelty lies only in chronic hypoxia influences exerted by this signaling. Authors definitely performed commendable in vivo studies to explore the role of the following in chronic hypoxia-induced PH: (i) genetic (smooth muscle cell-specific RyR2^{-/-} mice) and pharmacological (TTC) inhibition of RyR2, (ii) FKBP12.6/RyR2 complex stabilizer S107, and (iii) lentiviral shRNA-mediated SM-specific RISP knockdown. Thus, I would like the authors to rewrite the discussion that their novelty is based only on exploring this pathway in chronic hypoxia-induced pulmonary vascular remodeling and PH.

5. Author suggest that NF-κB/cyclinD1 pathway mediates RyR2-associated pulmonary vascular remodeling. However, no experiments were performed in addressing this aspect. Authors need to perform in vivo experiments treating RyR2 knockout mice with or without PDTTC, and evaluate the influence on pulmonary vascular remodeling, hemodynamics and right heart function.

Reviewer #4:

Remarks to the Author:

The paper by Mei et al uses multiple lines of experiments in pulmonary arterial smooth muscle cells (PAMCs) to advance the notion that chronic hypoxia-induced ROS production by RISP leads to RyR2 oxidation and FKBP12.6 dissociation, which in turn increases intracellular Ca leak leading to NF-κB and cyclin D1 activation, finally resulting in pulmonary hypertension.

The rationale for the experiments, the interpretation of results and the subsequent conclusions are logically laid out; however, as the bulk of the experiments is based on shaky or highly controversial premises (which is not even mentioned in the paper), it is very difficult to ascertain the validity of the final conclusions of the study (Fig. 7M). Specific concerns are as follows:

1. RyR2/FKBP12.6 dissociation is central to the development of pulmonary hypertension. The functional consequences of FKBP12.6 dissociation from RyR2 are not as clear cut as the authors state, with many reports (especially those from the Bers, Wayne Chen, and even Fleischer groups) finding

marginal effects on RyR2 function. Still, even if all those groups were incorrect (and to be fair their results were obtained in cardiomyocytes, not in PSMCs), then one would expect that the FKBP12.6 KO used here, which already has RyR2 "naked" channels, would exhibit rampant Ca leak and all the downstream effects that lead to PH. Instead, there is no apparent phenotype, and the authors' explanation that no effects are detected because upstream activation of RyR2 is required, is not congruent with their model. If RyR2 oxidation is required to dissociate FKBP12.6, but FKBP12.6 dissociation produces no effect, then the pathological event is RyR2 oxidation, only. Thus, authors statement that their primary mechanism for PH is "RyR2/FKBP12.6 remodeling" is not logically supported.

2. If RyR2 oxidation but not FKBP12.6 dissociation are required for PH, then the beneficial effects of S107 (so remarkably successful in all experiments, Fig. 5) cannot be explained based on the model of Fig 7M. Authors appear too focused on the controversial RyR2/FKBP12.6 interaction, excluding the possibility of other potential S107 interactions. After all, S107, like its parent molecule K201, is a benzothiazepine derivative proven to display pleiotropic effects. There is not even discussion on this matter.

3. FKBP12.6 is an immunophilin with proven targets outside the RyR2. Its role in metabolism, immunoregulation, transcription, is classical and remains more important than its potential and controversial role in RyR2 Ca regulation. The relative abundance of FKBP12.6 compared with RyR2, suggested by Western blots of Figs. 4&5, indicates additional roles for FKBP12.6 outside its potential effect on RyR2. Authors provide no measurement of RyR2/FKBP12.6 stoichiometry, or binding specificity, to adequately support model of Fig. 7M. Furthermore, there is no indication of the specificity of the FKBP12.6 antibody, i.e., is it cross-reacting with other FKBP12.x isoforms?

4. [3H]Ryanodine binding data is of limited value when not paired with accurate measurement of RyR2 density for the samples used in the binding assays. This is because [3H]ryanodine binding is the product of RyR2 activity x number of RyR2 channels. None of the binding curves were accompanied by proper control of RyR2 density (usually determined by Western blots, for example). Furthermore, there is no indication on how authors could obtain a preparation containing 10-15 pmols of RyR2 channels/mg protein (!). This is ~10-fold higher than RyR2 density in SR-enriched cardiac microsomes, and it simply appears too high a density in these PSMCs without several purification steps (not described). Please check these numbers.

Responses to Reviewers' Comments

We extremely appreciate all reviewers for providing excellent comments. Accordingly, we have meticulously addressed each of the comments by making appropriate text changes and/or including new experimental data, as described below. The major changes in the text have been written in red for your convenience. Overall, the quality of the manuscript has been substantially improved.

Reviewer #2 (Remarks to the Author):

In the current manuscript by Mei et al., entitled “Rieske iron-sulfur protein induces remodeling of FKBP12.6/RyR2 complex and subsequent pulmonary hypertension through NF- κ B/cyclin D1 pathway”, the authors investigate pathogenic links between Ca²⁺ signaling and its downstream signaling proteins and transcription factors in the development of pulmonary hypertension. A rise in intracellular [Ca²⁺]_i ([Ca²⁺]_i) in PASMC is a major trigger for pulmonary vasoconstriction and important stimulus for pulmonary vascular remodeling in patients with pulmonary arterial hypertension and animals with experimental pulmonary hypertension. The results from this study demonstrate a very interesting link between the RISP/ROS-mediated interaction of RyR2 and /FKBP12.6 with the Ca²⁺-sensitive NF- κ B/cyclin D1 signaling in PASMC proliferation and pulmonary vascular remodeling. The data presented in the study are novel and significant, and of high quality. The evidence for the novel link is compelling given the combined use of various techniques and in vitro/in vivo models in the study. I only have a few minor concerns that may need authors' attention to improve the manuscript.

Minor comments:

1. Please include molecular-weight size marker in Western blot images (so to allow readers to judge or estimate the size of target bands) shown in Figures 1(D), 4(A-B and F), 5(C), 6(A, C, D), 7(A, D, E, G) and S1 (A).

Thank you very much for the excellent suggestions. Accordingly, we have included the molecular-weight markers for all the relevant figures in our revised manuscript.

2. Although the study focuses on the RyR2-mediated Ca²⁺ signaling, it is important to show whether caffeine (or ryanodine)-mediated Ca²⁺ transients are affected by IP₃ and whether other RyRs and IP₃Rs are expressed in the cells used in the study.

This is a very interesting comment. In fact, our previous reports have shown that the caffeine-mediated Ca²⁺ transients are not affected by IP₃ in PASMCs, in which following caffeine (10 mM)-induced Ca²⁺ release, introduction of IP₃ (100 μ M) fails to induce further Ca²⁺ release (Zheng et al, J Gen Physiol, 2005, 125:427-440).

We have found that all three subtypes of RyRs (RyR1, RyR2 and RyR3) are expressed in PASMCs, but only RyR2 protein is highly present (Zheng et al, Cell Calcium, 2004, 35:345-355). Our findings have been confirmed by other investigators (Yang et al, Am J Physiol Lung Cell Mol Physiol 2005, 289:L338–L348). A series of our in-vitro studies indicate that all three RyR subtypes are involved in hypoxia-induced Ca²⁺ and contractile responses in PASMCs, but RyR2 is a very important player (Zheng et al, J Gen Physiol, 2005, 125:427-440; Li et al, Pflugers Arch - Eur J Physiol, 2009, 457:771–783; Liao et al, Antioxid Redox Signal, 2010, 14:37–47).

As we have reported, all three IP₃R subtypes (IP₃R1, IP₃R2 and IP₃R3) are expressed in PASMCs (Zheng et al, Cell Calcium, 2004, 35:345-355; Yadav et al, Am J Physiol Lung Cell Mol Physiol, 2018, 314:L724-L735); however, functional investigations indicate that only IP₃R1 may play an important role in hypoxic Ca²⁺ and contractile responses (Yadav et al, Am J Physiol Lung Cell Mol Physiol, 2018, 314:L724-L735).

3. It would be informative to discuss whether the RISP/ROS-mediated effect on RyR2/FKBP12.6 interaction (or dissociation) is unique to hypoxia-induced mitochondrial ROS production and hypoxia-induced pulmonary vasoconstriction/vascular remodeling. Cytoplasmic ROS can be increased by many extracellular and intracellular stimuli.

In response to this great comment, we have discussed the important role of RISP-mediated ROS in RyR2/FKBP12.6 interaction. Our series investigations have unveiled that hypoxia causes an increased generation in cytoplasmic ROS at two major sources such as mitochondria and NADPH oxidase (NOX) in PSMCs, and mitochondria are the primary and more important source; as such, ROS generated in mitochondria enter the cytosol, activate protein kinase C- ϵ (PKC ϵ), activate NOX, and then induce further ROS generation, a process termed ROS-induced ROS generation (RIRG), thereby contributing to hypoxia-induced Ca²⁺ and contractile response (Rathore et al, *Biochem Biophys Res Commun*, 2006, 351:784–790; Wang et al, *Free Radic Biol Med*, 2007, 42:642–653; Rathore et al, *Free Radic Biol Med*, 2008, 45:1223–1231). We have further demonstrated that hypoxic ROS generation in mitochondria of PSMCs primarily occurs at complex III, at which RISP plays an essential role (Korde et al, *Free Radic Biol Med*, 2011, 50:945–952). The essential role of RISP has been confirmed by other investigators (Waypa et al, *Am J Respir Crit Care Med*, 2013, 187:424–432).

Our in-vitro studies have revealed that RISP-mediated mitochondrial ROS are likely to be important for hypoxia-evoked dissociation of FKBP12.6 from RyR2, intracellular Ca²⁺ increase, contractile response in PSMCs (Liao et al, *Antioxid Redox Signal*, 2010, 14:37–47; Yadav et al, *Am J Physiol Lung Cell Mol Physiol*, 2013, 304: L143–L151). In the current study, we have demonstrated that lentiviral shRNA-mediated SM-specific RISP knockdown in vivo diminishes hypoxia-caused mitochondrial ROS generation, RyR2/FKBP12.6 complex dissociation, RyR2 activation, and PH in mice. These new data have not only significantly extended our previous in-vitro observations to in-vivo findings, but also greatly enhanced our mechanistic understanding of the development of PH, particularly with respect to the impact on the role of RISP-dependent ROS in mediating RyR2/FKBP12.6 interaction.

Reviewer #3 (Remarks to the Author):

The manuscript by Mei et al. reports that the Rieske iron-sulfur protein (RISP) induces remodeling of FKBP12.6/RyR2 complex and subsequently pulmonary hypertension (PH) through NF- κ B/cyclin D1 pathway. They elegantly identified that smooth muscle cell (SMC)-specific RyR2 knockout (KO) restores altered Ca²⁺ signaling, inhibits increased nuclear factor (NF)- κ B and downstream cyclin D1 activation as well as cell proliferation, and prevents PA vasoconstriction, remodeling and hypertension in CH-induced PH mice. FKBP12.6 KO or FK506 treatment enhances the sensitivity to CH-induced PH, while treatment of S107 (a specific stabilizer of RyR2/FKBP12.6 complex) produces an opposite effect. Lentiviral shRNA-mediated SMC-specific RISP knockdown can diminish mitochondrial reactive oxygen species (ROS) generation, RyR2/FKBP12.6 complex dissociation, and RyR2 activation, thereby preventing PH in CH mice. The manuscript is very convincing, highly relevant and comprehensive although I believe there are mainly novelty issues that are required to be addressed.

Major:

1. The authors conclude that to their knowledge, this is the first study to genetically and conditionally deplete RyR2 gene in smooth muscle cells in vivo. This is entirely misleading, as there is a study exploring the role of RyR2 in elementary Ca²⁺ Signaling in pulmonary arteries, pulmonary vasoconstriction (HPV) and PH. This is intentionally misleading to support a novel mechanism and not necessary. Targeted deletion of the RyR2 gene resulted in the enhanced pulmonary artery pressure response to sustained hypoxia (See Kaßmann M et al. *J Am Heart Assoc*. 2019 May 7;8(9):e010090). The authors do show a very novel aspect that could be important in pulmonary vascular remodeling underlying PH.

Thank you very much for pointing out a newly published, important article. In fact, we should have made additional literature search before we submitted our manuscript, although we did when we started to write our paper. In specific response to your comment, we have cited this very interesting article and made appropriate discussions in our revised manuscript. In particular, we have discussed: (1) the importance of the pioneering work using SM-conditional RyR2 KO mice, (2) their focus on the effects of acute hypoxia and our focus on the effects of chronic hypoxia, and (3) their ex-vivo lung perfusion model and our in-vivo pulmonary hypertension model.

2. Similarly, existing reports already suggest that RISP is a very important player in hypoxic ROS and Ca²⁺ signaling in PASMCs (See Song et al. Adv Exp Med Biol. 2017;967:289-298). The authors have extended these in vitro findings to in vivo studies.

We fully agree with your comment. A series of our in-vitro studies indicate that RISP is a very important player in acute hypoxic ROS and Ca²⁺ signaling in PASMCs, as pointed out in our recent review article (Song et al, Adv Exp Med Biol, 2017, 967:289-298). The current study, we have not only significantly extended our previous in-vitro findings to new in-vivo data with respect to the important role of RISP in mediating hypoxic ROS and Ca²⁺ signaling in PASMCs, but also greatly expanded our current mechanistic understanding of the essential contribution of RISP to the development of PH, particularly on RISP-associated signaling pathways.

3. In addition, there is a study which explored the role of RyR2/FKBP12.6 complex and the influence of FK506 in the context of HPV. This study identified that there is neither effect of displacement of FKBP12.6 from RyR2 nor FK506, a drug which displaces FKBP12.6 from ryanodine receptor 2 (RyR2), had any effect on HPV (See Connolly et al. J Physiol. 2013 Sep 15;591(18):4473-98).

We extremely appreciate this great comment. We have respectively reviewed this interesting article and agree with the reviewer that it is highly relevant. This important work has been cited and discussed in the revised manuscript, in which we have also particularly discussed the difference of the effects of in-vitro acute hypoxia in isolated rat PAs presented in the report by Connolly et al (J Physiol, 2013, 591:4473-98) and the effects of in-vivo chronic hypoxia in mice in the current study.

4. Similarly, the influence of PTDC (targeting NfKB signaling) on chronic hypoxia induced PH is not novel (see Fan J et al. High Alt Med Biol. 2016 Mar;17(1):43-9).

Based all the above-mentioned points and published literature in the field of RyR2 and HPV, the novelty lies only in chronic hypoxia influences exerted by this signaling. Authors definitely performed commendable in vivo studies to explore the role of the following in chronic hypoxia-induced PH: (i) genetic (smooth muscle cell-specific RyR2^{-/-} mice) and pharmacological (TTC) inhibition of RyR2, (ii) FKBP12.6/RyR2 complex stabilizer S107, and (iii) lentiviral shRNA-mediated SM-specific RISP knockdown. Thus, I would like the authors to rewrite the discussion that their novelty is based only on exploring this pathway in chronic hypoxia-induced pulmonary vascular remodeling and PH.

You are right, the influence of PTDC on hypoxia-induced PH has been reported by Fan J et al (High Alt Med Biol, 2016, 17:43-9). In addition to inclusion of this report, we have also followed your very commendable comment to rewrite the discussion that the major novelty of our work is based only on exploring the role of NF-κB as a downstream signaling molecule in mediating the important function of RyR2 in PH.

5. Author suggest that NF-κB/cyclinD1 pathway mediates RyR2-associated pulmonary vascular remodeling. However, no experiments were performed in addressing this aspect. Authors need to perform in vivo experiments treating RyR2 knockout mice with or without PDTC, and evaluate the influence on pulmonary vascular remodeling, hemodynamics and right heart function.

To follow this excellent suggestion, we have performed new in-vivo experiments treating RyR2 KO mice with or without PDTC. The results have been added in Figure 7 and indicate that PDTC treatment shows similar inhibitory effects on hypoxia-induced PH (right ventricular pressure and weight) in RyR2 KO and control mice. These new data further support our hypothesis that NF-κB serves as a downstream signaling molecule to mediate the role of RyR2 in PH; as such, RyR2 KO, PDTC treatment, and RyR2 KO/PDTC treatment all inhibit the development of PH to a similar extent.

Reviewer #4 (Remarks to the Author):

The paper by Mei et al uses multiple lines of experiments in pulmonary arterial smooth muscle cells (PASMCs) to advance the notion that chronic hypoxia-induced ROS production by RISP leads to RyR2 oxidation and FKBP12.6 dissociation, which in turn increases intracellular Ca leak leading to NF-κB and cyclin D1 activation, finally resulting in pulmonary hypertension.

The rationale for the experiments, the interpretation of results and the subsequent conclusions are logically laid out; however, as the bulk of the experiments is based on shaky or highly controversial premises (which is not even mentioned in the paper), it is very difficult to ascertain the validity of the final conclusions of the study (Fig. 7M). Specific concerns are as follows:

1. RyR2/FKBP12.6 dissociation is central to the development of pulmonary hypertension. The functional consequences of FKBP12.6 dissociation from RyR2 are not as clear cut as the authors state, with many reports (especially those from the Bers, Wayne Chen, and even Fleischer groups) finding marginal effects on RyR2 function. Still, even if all those groups were incorrect (and to be fair their results were obtained in cardiomyocytes, not in PSMCs), then one would expect that the FKBP12.6 KO used here, which already has RyR2 "naked" channels, would exhibit rampant Ca leak and all the downstream effects that lead to PH. Instead, there is no apparent phenotype, and the authors' explanation that no effects are detected because upstream activation of RyR2 is required, is not congruent with their model. If RyR2 oxidation is required to dissociate FKBP12.6, but FKBP12.6 dissociation produces no effect, then the pathological event is RyR2 oxidation, only. Thus, authors statement that their primary mechanism for PH is "RyR2/FKBP12.6 remodeling" is not logically supported.

Thank you very much for this excellent comment. Indeed, a report from Dr. Bers' group indicate that only a small portion of endogenous RyR2s are associated with FKBP12.6 cardiac myocytes; despite this, however, FKBP12.6 inhibits basal RyR2 activity (Gu et al, *Circ Res*. 2010, 106:1743). With respect to the binding site of RyR2 to FKBP12.6, studies have shown inconsistent results. Chen and colleagues (Masumiya et al, *J Biol Chem*, 2003, 278:3786-92; Zhang et al, *J Biol Chem* 2003, 278:14211-8) have identified the binding site of FKBP12.6 binds to RyR2 amino acids between 1815–1855, and isoleucine 2427 is not required. On the other hand, Marks and colleagues identified that isoleucine 2427 and proline residues 2428 are the binding site for FKBP12.6 (Marx et al, *Cell* 2000, 101:365-76). The finding that amino acids 1815–1855 are crucial for FKBP12.6 association with RyR2 has been supported by later reports by Zissimopoulos and Lai (Zissimopoulos & Lai, *J Biol Chem* 2005, 280:5475-85) and others (Yan et al, *Nature*, 2014, 517:50–55; Efremov et al, *Nature* 2015, 517:39-43; Oda et al, *J Mol Cell Cardiol*, 2015, 85:240–248; des Georges et al, 2016, *Cell* 167:145–157).

FKBP12.6 KO mice we generated on the 129SvEv background show the increased Ca²⁺ release in myocardiocytes; moreover, cardiac hypertrophy occurs in male, but not female KO mice (Xin et al, *Nature* 2002, 416:334-8). Interestingly, FKBP12.6 KO mice on a DBA genetic background have the increased Ca²⁺ release, but normal hearts; these mice have exercise-induced sudden cardiac death (Wehrens et al, *Cell* 2003, 113:829-40). In support of the role of FKBP12.6, overexpression of this protein inhibits Ca²⁺ release in cardiac cells (Prestle et al, *Circ Res* 2001, 88:188-94; Gomez et al, *Am J Physiol Heart Circ Physiol* 2004, 287:H1987-93; Loughrey et al, *J Physiol* 2004, 556:919-34). Taken together, FKBP12.6 binds to RyR2 and functions as an endogenous channel inhibitor (stabilizer) in cardiac cells.

As you stated, the findings could be different in cardiac cells and PSMCs. However, the inhibition of RyR2 by FKBP12.6 has been reported in PSMCs, in which chemical or genetical removal of FKBP12.6 (FK506 treatment or KO) significantly increases Ca²⁺ release (Zheng YM et al, *Cell Calcium*, 2004, 35: 345-355; Liao B et al, *Antioxid Redox Signal*, 2011, 14: 37-47). A similar function of FKBP12.6 has also been observed in airway SMCs (Wang YX et al, *Am J Physiol Cell Physiol*. 2004, 286: C538-46; Du et al, *Am J Respir Cell Mol Biol*, 2014, 50: 398-408). These reports indicate that the role of FKBP12.6 in the regulation of Ca²⁺ release in PSMCs is overall comparable to that in cardiac cells.

In the current study, we have found that genetic and chemical FKBP12.6 removal both promote, whereas use of the small molecule S107 to stabilize FKBP12.6, RyR antagonist TTC to inhibit the channel activity and RyR2 KO mice to annul the channel function, all block HYPOXIA-induced PH. We fully agree with the reviewer's view that FKBP12.6 removal cannot induce PH on its own; on the other hand, the current data with previous findings demonstrate that hypoxia can cause FKBP12.6 removal and then enhance RyR2 activity, which may synergistically act with hypoxic oxidation-induced activation of RyR2 to produce PH; accordingly, both FKBP12.6 and RyR2 both contribute to the development of hypoxic PH. Otherwise, FKBP12.6 removal and stabilization (inhibition) do not produce effects. As for use of the term "RyR2/FKBP12.6 remodeling", we would be willing to make a change if you could suggest.

2. If RyR2 oxidation but not FKBP12.6 dissociation are required for PH, then the beneficial effects of S107 (so remarkably successful in all experiments, Fig. 5) cannot be explained based on the model of Fig 7M. Authors appear too focused on the controversial RyR2/FKBP12.6 interaction, excluding the possibility of other potential S107 interactions. After all, S107, like its parent molecule K201, is a benzothiazepine derivative proven to display pleiotropic effects. There is not even discussion on this matter.

We extremely appreciate this interesting point. As discussed above, our data indicate both RyR2 oxidation and FKBP12.6 removal are required the hypoxia-induced (Group 3) PH. Indeed, RyR2/FKBP12.6 interaction has been extensively studied in cardiac cells, and the findings are not always consistent. In contrast, both molecule interaction has not been extensively investigated in SMCs, but the results are consistent and reveal that their interaction may play an important role in the regulation of intracellular Ca^{2+} and contraction in PSMCs and other SMCs. Because of the limited information available, we believe that our current work is of particular impact in the field.

In specific response to your comment on the pleiotropic effects of S107, we have added discussions in the manuscript on this issue. Thank you very much again.

3. FKBP12.6 is an immunophilin with proven targets outside the RyR2. Its role in metabolism, immunoregulation, transcription, is classical and remains more important than its potential and controversial role in RyR2 Ca regulation. The relative abundance of FKBP12.6 compared with RyR2, suggested by Western blots of Figs. 4&5, indicates additional roles for FKBP12.6 outside its potential effect on RyR2. Authors provide no measurement of RyR2/FKBP12.6 stoichiometry, or binding specificity, to adequately support model of Fig. 7M. Furthermore, there is no indication of the specificity of the FKBP12.6 antibody, i.e., is it cross-reacting with other FKBP12.x isoforms?

You are right, FKBP12.6 is involved in multiple cellular responses. A study using FKBP12.6 KO mice has shown that FKBP12.6 is associated with glucose metabolism; the role of FKBP12.6 is believed to be implemented by its interaction with RyR2 in pancreatic islets (Chen et al, FASEB J. 2010, 24:357-63). Similarly, investigations of FKBP12.6 KO mice have also reported that this protein may mediate learning processes as a result of the increased RyR2 activity due to the its dissociation (Yuan et al, Sci Rep, 2016, 6:21087; Gant et al, J Neurosci, 2018, 38:1030-1041). Pharmacological studies have found that FKBP12.6/RyR2 dissociation is possibly involved in apoptosis of skin fibroblasts (Tang et al, Cell Physiol Biochem, 2016, 39:1965-1976). These results are consistent with the role of FKBP12.6 in hypoxic contractile responses PSMCs as described in our earlier publications (Zheng YM et al, Cell Calcium, 2004, 35: 345-355; Liao B et al, Antioxid Redox Signal, 2011, 14: 37-47), neurotransmitter-induced contraction in airway SMCs as reported by us (Wang YX et al, Am J Physiol Cell Physiol. 2004, 286: C538-46) and others (Du et al, Am J Respir Cell Mol Biol, 2014, 50: 398-408), and hypoxia-induced PH in vivo in the current study with multiple complementary approaches.

Overall, biochemical and structural studies have shown the specificity of RyR2/FKBP12.6 binding is high at least in cardiac cells (Xin et al, J Biol Chem, 1999, 274:15315–15319; Marx et al, Cell 2000, 101:365-76; Masumiya et al, J Biol Chem, 2003, 278:3786-92; Zhang et al, J Biol Chem, 2003, 278:14211-8; Gu et al, Circ Res. 2010, 106:1743; Zissimopoulos S, Lai FA. J Biol Chem, 2005, 280:5475-5485; Yan et al, Nature, 2014, 517:50–55; Efremov et al, Nature, 2015, 517:39-43; Oda et al, J Mol Cell Cardiol, 2015, 85:240–248; des Georges et al, 2016, Cell 167:145–157). As we reported, FKBP12.6 protein, but not FKBP12, is highly expressed in PSMCs; glutathione S-transferase (GST)-FKBP12.6 fusion protein is able to associate with RyR2, but GST-FKBP12 protein is not (Zheng et al, Cell Calcium, 2004, 35:345–355). We have made similar findings in airway SMCs (Wang et al, Am J Physiol Cell Physiol, 2004, 286:C538–C546). These reports reveal that FKBP12.6 specifically associates with RyR2 in PSMCs.

4. [3H]Ryanodine binding data is of limited value when not paired with accurate measurement of RyR2 density for the samples used in the binding assays. This is because [3H]ryanodine binding is the product of RyR2 activity x number of RyR2 channels. None of the binding curves were accompanied by proper control of RyR2 density (usually determined by Western blots, for example). Furthermore, there is no indication on how authors could obtain a preparation containing 10-15 pmols of RyR2 channels/mg protein (!). This is ~10-fold higher

than RyR2 density in SR-enriched cardiac microsomes, and it simply appears too high a density in these PSMCs without several purification steps (not described). Please check these numbers.

In general, we respectively agree with your comment. In fact, we interpret [³H] ryanodine binding data with the findings from experiments using RyR2 KO mice, RyR antagonist, FRET and RyR2/FKBP12.6 ratio measurements; thus, the relevant statements and conclusions are made by integrative considerations of all experimental data.

We have provided more details about the sample preparations in the Methods section. Please note that we have made a correction, in which a coefficient value of 20 should be applied to B_{\max} . The corrected, calculated values have been included in the revised manuscript. Thank you very much again for this and all other thoughtful comments.

Reviewers' Comments:

Reviewer #2:

Remarks to the Author:

The authors have adequately and appropriately addressed my concerns and questions. No further comments.

Reviewer #3:

Remarks to the Author:

Comment

I am mostly satisfied with the additional experiments and responses that were provided by the authors to my previous comments however; I would like authors to give more details on lentivirus mediated RISP shRNA delivery in the methods section. It is not clear how much volume was given, is it 3mL as stated in the methods? It is also not clear what was virus titer administered to a mouse? Did authors indeed use jugular vein to inject the virus, if yes, they should provide more details of jugular vein injection as it is invasive procedure. Two references they have cited for this method used tail veins for intravenous injection. In addition, they should provide if there was single injection (at what time point of the experiment – before or in the middle of hypoxia exposure?) or were few injections.

Reviewer #4:

Remarks to the Author:

Authors have addressed all my concerns, but only a few have been resolved. My main concern with the data continues to be that some crucial conclusions do not agree well with the model presented in the last figure, and therefore with the main conclusions of the paper. Essentially, authors addressed my concerns by just repeating the same statements, but in different words, and also by using selected references that support their conclusions, but ignoring others that don't.

The controversy on the RyR2/FKBP12.6 interaction is on the EFFECT of FKBP12.6, not on its binding site. I suggest that authors delete the section where they mention the mapping of RyR amino acid residues involved in FKBP12.x binding because this discussion is irrelevant for this paper. To be fair, they should select other references that demonstrate that FKBP12.6 binding has NO or very limited effect on RyR2 function. In fact, the Bers paper that they cite as positive demonstration of the inhibition of RyR2 activity by FKBP12.6 is instead a demonstration that PHYSIOLOGICAL levels of FKBP12.6 do nothing to RyR2 activity (please refer again to this paper and cite it properly). Other papers by Chen, Fleischer, Valdivia, Houser, etc. do NOT support the role of FKBP12.6 postulated by Marks (which is the model used by authors).

It is important to note that I'm not arguing that data in this paper is inaccurate. My concern is that authors have chosen to interpret it based on a hypothetical scheme that is highly controversial and that they portray as an established fact. The last part is the inaccurate part of the paper. All I'm suggesting is that authors clearly state that the mechanism of RyR2 inhibition by FKBP12.6 is not clearly established and that FKBP12.6 and FKBP12.0 have a variety of effects on molecules other than the RyR2 (and thus, the results of the paper may be interpreted differently). These statements should give readers a more balanced view of the present state of affairs with regards to the "RyR2/FKBP12.6 interaction".

Reviewers' comments:

Reviewer #2 (Remarks to the Author):

The authors have adequately and appropriately addressed my concerns and questions. No further comments.

Answer: Thank you very much for the excellent comments again!

Reviewer #3 (Remarks to the Author):

Comment

I am mostly satisfied with the additional experiments and responses that were provided by the authors to my previous comments however; I would like authors to give more details on lentivirus mediated RISP shRNA delivery in the methods section. It is not clear how much volume was given, is it 3mL as stated in the methods? It is also not clear what was virus titer administered to a mouse? Did authors indeed use jugular vein to inject the virus, if yes, they should provide more details of jugular vein injection as it is invasive procedure. Two references they have cited for this method used tail veins for intravenous injection. In addition, they should provide if there was single injection (at what time point of the experiment – before or in the middle of hypoxia exposure?) or were few injections.

Answer: Again, we greatly appreciate your constructive suggestion. The more details on lentivirus mediated RISP shRNA delivery have been provided in the methods section, in which a 300- μ l solution containing either lentiviral RISP or scrambled shRNAs was administrated to each individual mouse via an intrajugular vein. Viral concentration was 3×10^8 transducing unit ml. All mice were received a single injection; after 24 hours, mice were exposed to hypoxia or normoxia for 3 weeks.

Reviewer #4 (Remarks to the Author):

Authors have addressed all my concerns, but only a few have been resolved. My main concern with the data continues to be that some crucial conclusions do not agree well with the model presented in the last figure, and therefore with the main conclusions of the paper. Essentially, authors addressed my concerns by just repeating the same statements, but in different words, and also by using selected references that support their conclusions, but ignoring others that don't.

The controversy on the RyR2/FKBP12.6 interaction is on the EFFECT of FKBP12.6, not on its binding site. I suggest that authors delete the section where they mention the mapping of RyR amino acid residues involved in FKBP12.x binding because this discussion is irrelevant for this paper. To be fair, they should select other references that demonstrate that FKBP12.6 binding has NO or very limited effect on RyR2 function. In fact, the Bers paper that they cite as positive demonstration of the inhibition of RyR2 activity by FKBP12.6 is instead a demonstration that PHYSIOLOGICAL levels of FKBP12.6 do nothing to RyR2 activity (please refer again to this

paper and cite it properly). Other papers by Chen, Fleischer, Valdivia, Houser, etc. do NOT support the role of FKBP12.6 postulated by Marks (which is the model used by authors).

It is important to note that I'm not arguing that data in this paper is inaccurate. My concern is that authors have chosen to interpret it based on a hypothetical scheme that is highly controversial and that they portray as an established fact. The last part is the inaccurate part of the paper. All I'm suggesting is that authors clearly state that the mechanism of RyR2 inhibition by FKBP12.6 is not clearly established and that FKBP12.6 and FKBP12.0 have a variety of effects on molecules other than the RyR2 (and thus, the results of the paper may be interpreted differently). These statements should give readers a more balanced view of the present state of affairs with regards to the "RyR2/FKBP12.6 interaction".

Answer: We thank you very much for your comments and have tried to address your concerns more directly in this response and in the text of the manuscript. Accordingly, we have deleted the discussion of RyR2/FKBP12 binding sites and attempted to make our interpretation more neutral, balancing different points of view. In particular, the discussion concerning a report by Bers and associates has been substantially changed following your suggestions, indicating that the physiological level of FKBP12.6 might not affect RyR2 activity. This view is different from a series of studies by Marks and colleagues, in which they showed that FKBP12.6 mediates several cardiac diseases due to the dysfunctional mechanism of RyR2 inhibition by FKBP12.6. More citations have been added to support the findings by Bers' group. Nevertheless, we clearly state that the mechanism of RyR2 inhibition by FKBP12.6 postulated by Marks and colleagues is not clearly established and FKBP12.6 and FKBP12.0 may have a variety of effects on molecules other than the RyR2 in cardiac myocytes.

There are minimal studies examining these mechanisms in the pulmonary vasculature which may yield different findings. Our previous findings and current studies using FKBP12.6 KO mice, FKBP12.6 stabilizer, RyR2 KO mice and RyR blocker provide strong evidence that the mechanism of RyR2 inhibition by FKBP12.6 mediates hypoxia-induced pulmonary vasoconstriction, remodeling and hypertension. A few reports suggest that this mechanism may function in airway SMCs and pancreatic cells as well. However, it should be pointed that more future studies can substantially strengthen our view of the present state of affairs with regards to the RyR2/FKBP12.6 interaction in PSMCs.

Reviewers' Comments:

Reviewer #3:

Remarks to the Author:

The authors have adequately and appropriately addressed my concerns and questions. No further comments.

Reviewer #4:

Remarks to the Author:

Authors have addressed my comments, but not my concerns. Overall, the message of the manuscript with regards to the controversial FKBP12.6/RyR2 interaction changed almost nothing. However, as I stated before, I don't dispute the accuracy of these results and have no reason to doubt that the FKBP12.6/RyR2 interaction is different in the lungs compared to the heart. Therefore, I respect the authors' evident bias in interpreting their results based on the heart model. I have no further comments.

REVIEWERS' COMMENTS:

Reviewer #3 (Remarks to the Author):

The authors have adequately and appropriately addressed my concerns and questions. No further comments.

Answer: Again, we thank you very much for your excellent comments and efforts.

Reviewer #4 (Remarks to the Author):

Authors have addressed my comments, but not my concerns. Overall, the message of the manuscript with regards to the controversial FKBP12.6/RyR2 interaction changed almost nothing. However, as I stated before, I don't dispute the accuracy of these results and have no reason to doubt that the FKBP12.6/RyR2 interaction is different in the lungs compared to the heart. Therefore, I respect the authors' evident bias in interpreting their results based on the heart model. **I have no further comments.**

Answer: As you “have no further comments”, we have not made any significant changes in the current revised manuscript. Once more, we extremely appreciate your very valuable comments.